# ECLayr: A Fast and Robust Topological Layer via Euler Characteristic Curves

## Abstract

We introduce a flexible and computationally efficient topological layer for general deep learning architectures, built upon the Euler Characteristic Curve. Unlike existing approaches that rely on computationally intensive persistent homology, our method bypasses this bottleneck while retaining essential topological information across diverse data modalities. To enable complete end-to-end training, we develop a novel backpropagation scheme that improves computation and mitigates vanishing gradient issues. We go on to provide stability analysis, establishing stability guarantees for the proposed layer in the presence of noise and outliers. We integrate the proposed layer into topological autoencoders to enhance representation learning through topological signals. We further demonstrate the effectiveness of our approach through classification experiments on a variety of datasets, including high-dimensional settings where persistent homology becomes computationally challenging.

## 1 Introduction

In recent years, machine learning communities have witnessed increasing efforts to incorporate Topological Data Analysis (TDA) into deep learning workflows, an emerging paradigm known as topological deep learning (Carlsson & Gabrielsson, 2020; Papamarkou et al., 2024). This approach integrates tools from TDA to exploit essential topological features that are often inaccessible to traditional deep learning methods, or to enhance data representation and model interpretability. *Persistent homology* (PH), a fundamental tool in TDA, captures multi-scale topological features of the underlying data structure by tracking the birth and death of homology features, thereby producing topological summaries such as persistence diagrams or barcodes. Given that PH is a multiset by nature, a number of strategies have been proposed to transform these PH-based topological summaries into alternative representations that are more suitable for subsequent machine learning tasks (e.g., Bubenik et al., 2015; Chazal et al., 2014b; Adams et al., 2017, see, for example, Hensel et al. (2021) for a review).

Recent efforts have shed light on the possibility of incorporating PH-based topological summaries as input features for neural networks, enhancing their ability to learn from the intrinsic geometric structure of the data. Hofer et al. (2017); Carrière et al. (2020) introduced parametrized topological layers designed to learn vector embeddings of persistence diagrams, although the differentiability required for gradient-based optimization was limited to the layer parameterization and did not extend to layer inputs. Gameiro et al. (2016); Chen et al. (2019); Gabrielsson et al. (2020); Carriere et al. (2021); Leygonie et al. (2022); Carriere et al. (2024) explored the differentiability of PH-based functions or losses, highlighting the potential of integrating topological insights into deep learning frameworks. Kim et al. (2020) were the first to propose a generic differentiable topological layer that allows backpropagation, offering flexibility in its integration within arbitrary network architectures.

Despite its popularity, PH often incurs substantial computational overhead, rendering it impractical for large-scale deep learning applications. Its time complexity generally scales poorly with the size and dimensionality of the data; for instance, the standard PH algorithm has a time complexity of $O(N^3)$, where $N$ denotes the number of simplices (Otter et al., 2017). As a result, there has recently been a growing need for alternative features capable of capturing topological information in a computationally efficient manner. The *Euler*

*Characteristic Curve* (ECC) is one such feature that can be computed without the need for PH computation. ECC-based descriptors have recently gained attention for their ability to significantly improve computational efficiency (Richardson & Werman, 2014; Beltramo et al., 2021; Chen et al., 2022; Dłotko & Gurnari, 2023; Hacquard & Lebovici, 2023; Malott & Wilsey, 2023; Jiang et al., 2023; Laky & Zavala, 2024). However, prior works have primarily employed these descriptors in a static fashion, limiting their use to traditional feature engineering frameworks.

A more recent advancement in this area is the Differentiable Euler Characteristic Transform (DECT) (Röell & Rieck, 2024), which is a differentiable topological layer built upon the Euler Characteristic Transform (ECT), i.e., a collection of ECCs computed along multiple directions (Turner et al., 2014). While promising, the existing DECT framework is specifically tailored to geometric data such as graphs and meshes. Consequently, it does not generalize naturally to other data modalities, such as images or voxels. Moreover, its differentiability relies on a sigmoid-based approximation, which incurs considerable computational overhead and may lead to gradient inconsistency or vanishing effects, issues that will be further examined in Section 4. In addition, such DECT-based learning methods remain in an early stage of development, and a rigorous analytical investigation of their theoretical properties has yet to be conducted.

**Contribution.** We propose a novel ECC-based topological layer, *ECLayr*, and make six main contributions. First, we demonstrate both theoretically and empirically that ECLayr substantially improves computational efficiency by eliminating the need for PH computation. Second, in contrast to prior ECT-based studies such as Röell & Rieck (2024), ECLayr accommodates generic differentiable filtrations, offering a flexible and unified framework that extends naturally to diverse data modalities, including images and voxels. Third, we introduce an efficient backpropagation strategy that resolves the shortcomings of sigmoid-based differentiability used in earlier approaches. Fourth, we present a rigorous stability analysis characterizing the stability bounds of the proposed layer under noisy perturbations in the input. Fifth, we showcase the versatility of ECLayr through its application in topological autoencoders. Finally, our classification experiments reveal that, when the underlying topology is relatively simple, ECLayr achieves performance comparable to state-of-the-art PH-based layers while being significantly faster. Further experiments show that ECLayr is capable of effectively capturing topological information even in complex, high-dimensional settings, where PH becomes computationally prohibitive.

## 2 Preliminaries

This section provides a brief overview of the essential tools in TDA used throughout the development, as well as some notations. For further information, see, for example, Hatcher (2002); Kaczynski et al. (2004); Edelsbrunner & Harer (2010); Chazal & Michel (2021).

### 2.1 Definitions

**Simplex and Simplicial Complex.** A *k-simplex* is the convex hull of $k+1$ affinely independent points $u_0, \ldots, u_k$ in $\mathbb{R}^d$, i.e., $\sigma_k = \text{conv}\{u_0, \ldots, u_k\}$ (e.g., 0-simplex: vertex, 1-simplex: edge, 2-simplex: triangle, etc.). The *dimension* of $\sigma_k$ is $k$. $\tau$ is a *face* of $\sigma_k$ if it is a convex hull constructed from any non-empty subset of the $k+1$ points in $\sigma_k$. A *simplicial complex* $K$ is a finite collection of simplices such that (i) the face of any simplex in $K$ is also in $K$, and (ii) the intersection of two simplices in $K$ is either empty or a face of both. Commonly used simplicial complexes include the Vietoris-Rips complex and the Alpha complex (see Appendix C.1 for detailed definitions).

**Cubical Complex.** A cubical complex is an analogue of a simplicial complex that consists of $k$-cubes (e.g., vertices, edges, squares, cubes, etc.), and it provides a suitable framework for analyzing data that naturally align with a grid structure, such as digital images. For $l \in \mathbb{Z}$, an *elementary interval* is an interval of the form $I = [l, l+1]$ or $I = [l, l]$, where the former interval is called *nondegenerate* and the latter *degenerate*. An *elementary cube* is the finite product of elementary intervals, i.e., $Q = I_1 \times I_2 \times \cdots \times I_n$. The *dimension* of $Q$ is the number of nondegenerate elementary intervals in the product. Given that $P$ and $Q$ are both elementary cubes, $P$ is a *face* of $Q$ if $P \subset Q$. A *cubical complex* $K$ is a finite collection of elementary

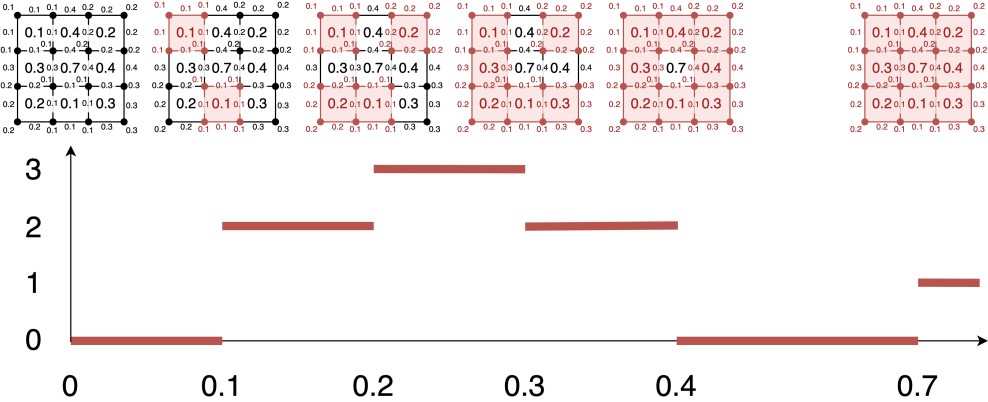

Figure 1: Computation of ECC using sublevel set filtration on a filtered cubical complex.

cubes such that the face of any cube in $K$ is also in $K$, and a *filtered cubical complex* can be constructed by assigning a filtration value to each cube (see Appendix C.2).

**Filtration.** A *filtration* $\mathcal{F} = \{K(t) \subset K \mid t \in \mathbb{R}\}$ is a collection of nested subcomplexes that satisfy $K(t_1) \subset K(t_2)$ whenever $t_1 \leq t_2$. A typical way of constructing a filtration is to use a monotonic filtration function $f : K \rightarrow \mathbb{R}$, where $f$ is monotonic in the sense that $f(\tau) \leq f(\sigma)$ whenever $\tau$ is a face of $\sigma$. By defining $K(t) \coloneqq f^{-1}(-\infty, t]$, we have $K(t_1) \subset K(t_2)$ whenever $t_1 \leq t_2$.

**Persistent Homology and Persistence Diagram.** *Persistent homology* is a multiscale approach to capture the topological features of a complex $K$, and can be represented as a persistence diagram. For a filtration $\mathcal{F}$ and for each nonnegative $k$, we keep track of when $k$-dimensional homological features (e.g., 0-dimension: connected component, 1-dimension: loop, 2-dimension: cavity, etc.) appear and disappear in the filtration. If a homological feature $\alpha_i$ appears at filtration value $b_i$ and disappears at $d_i$, we say that $\alpha_i$ is born at $b_i$ and dies at $d_i$. Treating these pairs $(b_i, d_i)$ as points in the plane $\mathbb{R}_*^2 \coloneqq \{(b, d) \in (\mathbb{R} \cup \infty)^2 : d > b\}$ yields the *persistence diagram*, a finite multiset defined as

$$\mathcal{D} = \{(b_i, d_i) : (b_i, d_i) \in \mathbb{R}_*^2\}.$$

**Euler Characteristic.** The Euler characteristic is a topological invariant that provides a single number summary of the underlying topological[1] structure. Given a complex $K$, the *Euler characteristic* is an alternating sum of the number of $k$-cells (e.g., $k$-simplices or $k$-cubes) in $K$. It can equivalently be expressed as an alternating sum of Betti numbers:

$$\chi(K) = \sum_{k=0}^{\infty} (-1)^k |K^k| = \sum_{k=0}^{\infty} (-1)^k \beta_k, \tag{1}$$

where $K^k$ is the set of $k$-cells in $K$, $|K^k|$ is its cardinality, and $\beta_k$ is the $k$-th Betti number of $K$.

**Euler Characteristic Curve.** The *Euler Characteristic Curve* (ECC) $\mathcal{C} : \mathbb{R} \rightarrow \mathbb{R}$ is obtained by evaluating the Euler characteristic along a filtration, with the filtration value on the $x$-axis and the Euler characteristic of the corresponding subcomplex on the $y$-axis (see Figure 1). For $t \in \mathbb{R}$, the ECC is given as

$$\mathcal{C}(t) = \chi(K(t)), \tag{2}$$

where $K(t) = \{\sigma \in K : f(\sigma) \leq t\}$ is the subcomplex of $K$ at $t$ and $f(\sigma)$ is the filtration value of $\sigma$.

## 2.2 Trade-off between ECC- and PH-based Approaches

Although ECC offers an efficient means of capturing topological information, there exists an inherent trade-off between ECC- and PH-based approaches. In particular, ECC is generally less expressive than PH in

---

[1]Depending on the filtration, both PH and ECC can also capture geometric information.

---

**Algorithm 1:** Computation of ECC: $X \to \mathcal{E}$

---

**1** **Hyperparameters:** $T_{min}$, $T_{max}$, $v$

**2** **Input:** data $X$

**3** Choose a simplicial or cubical complex suitable for the input data and build a filtration.

**4** Set $tseq = \{t_1, \ldots, t_v\}$, an evenly-spaced grid of size $v$ where $t_1 = T_{min}$ and $t_v = T_{max}$.

**5** Initialize $\mathcal{E} = (0, \ldots, 0) \in \mathbb{R}^v$.

**6** **for** $\sigma \in K$ **do**

**7**    **if** $f(\sigma) > T_{max}$ **then**

**8**       continue

**9**    $i^* \leftarrow \arg\min_i\{t_i \in tseq \,|\, t_i > f(\sigma)\}$

**10**    $\mathcal{E}(i^*) \leftarrow \mathcal{E}(i^*) + (-1)^{\dim(\sigma)}$

**11** $\mathcal{E} \leftarrow \text{cumsum}(\mathcal{E})$

**12** **Output**: $\mathcal{E} \in \mathbb{R}^v$

---

capturing fine-grained topological features, as it aggregates information across all homology dimensions, whereas PH provides dimension-specific summaries. For example, consider point clouds sampled from (i) a single loop of radius $r$, and (ii) two disjoint loops of radius $r$. PH distinguishes the underlying structures via $(\beta_0, \beta_1) = (1, 1)$ versus $(2, 2)$, whereas ECC cannot since both have Euler characteristic $\chi = \beta_0 - \beta_1 = 0$. This efficiency-expressivity trade-off between ECC and PH yields distinct advantages for ECC- and PH-based approaches, motivating a characterization of the settings in which ECLayr may be preferable. First, ECLayr is primarily intended for applications where computational efficiency is prioritized over granular topological detail, particularly in high-dimensional or resource-constrained settings where PH becomes computationally prohibitive. In such settings, where PH is often not practically applicable, ECLayr provides a viable means of extracting topological information. Furthermore, ECLayr offers a computationally attractive alternative when the underlying topology is relatively simple, in the sense that distinct data structures are uniquely characterized by their aggregated Betti numbers. For instance, in the MNIST handwritten digits dataset, which we use as a benchmark in Section 6.3, all digits have a single connected component ($\beta_0 = 1$), so the primary discriminative signal arises from loop counts. Since ECC distinguishes digits with zero, one, and two loops (e.g., 1/0/8) via $\chi \in \{1, 0, -1\}$, ECLayr can be as informative as PH-based models in this setting while being orders of magnitude faster. In contrast, when differences in the Betti numbers across homology dimensions cancel in the alternating sum, as in the point cloud example above, PH-based models may be preferable. Overall, ECLayr is designed for efficiency-critical tasks, such as high-dimensional settings where PH is costly, and for applications that admit relatively simple topological structures.

## 3 Layer Construction

The construction of ECLayr involves two steps: (i) computing a vectorized approximation of ECC from input data, and (ii) passing the output of (i) through a differentiable map. To compute the vectorized approximation of ECC, we consider an alternative definition of the Euler characteristic. Given the subcomplex $K(t) \subset K$ at filtration value $t$, the Euler characteristic in equation 1 can be equivalently defined as

$$\chi(K(t)) = \sum_{k=0}^{\infty} (-1)^k \sum_{\sigma \in K^k} \mathbb{1}\left[f(\sigma) \leq t\right], \tag{3}$$

where $f(\sigma)$ is the filtration value of $\sigma$. The equivalence between equation 1 and equation 3 is straightforward, as the sum of the indicators in equation 3 equals the number of $k$-simplices in the subcomplex $K(t)$.

**Notations.** For notational simplicity, let $X$ denote the input, $\mathcal{E}$ the *vectorized* ECC, and $\mathcal{O}_\theta$ the output of our layer parameterized by $\theta$.

### 3.1  Computation of Vectorized ECC: $X \to \mathcal{E}$

To calculate the ECC from input data, we must first define a filtration by selecting an appropriate simplicial or cubical complex $K$ and a function $f : K \to \mathbb{R}$. This choice is often data- or problem-dependent. Upon constructing a filtration, we proceed to obtain the vectorized approximation of ECC based on equation 3. First, we set a closed interval $[T_{min}, T_{max}]$ and sample $v$ evenly-spaced grid points ranging from $T_{min}$ to $T_{max}$. We denote these discretized points as $tseq = \{t_1, \ldots, t_v\}$, where $t_1 = T_{min}$ and $t_v = T_{max}$. Our objective is to derive a vector $\mathcal{E}$ containing the Euler characteristics of each subcomplex $K(t_i)$, i.e., $\mathcal{E} = (\chi(K(t_1)), \ldots, \chi(K(t_v)))$. This vector serves as a finite sample approximation of the ECC function $\mathcal{C}$ defined in equation 2. To compute $\mathcal{E}$, we begin by initializing $\mathcal{E}$ as a zero vector of size $v$. Next, we iterate over all simplices $\sigma \in K$ and perform the following steps: (i) find $i^* := \arg\min_i\{t_i \in tseq \,|\, t_i > f(\sigma)\}$, which denotes the smallest grid point that is larger than the filtration value of $\sigma$, and (ii) add $(-1)^{\dim(\sigma)}$ to $\mathcal{E}(i^*)$. If $f(\sigma)$ exceeds the upper bound $T_{max}$ and $i^*$ cannot defined, we proceed to the subsequent simplex in the iteration. Once the iteration is terminated, we return the cumulative sum of $\mathcal{E}$ for each grid point. The resulting output $\mathcal{E}$ is a vector in $\mathbb{R}^v$. The overall procedure is summarized in Algorithm 1.

**Runtime.** Informally, the ECC computation in Algorithm 1 (steps 6–12) runs in $O(N + v)$ time, where $N$ denotes the number of simplices and $v$ the number of grid points. We formalize this statement in the following proposition.

**Proposition 3.1.** *For a random input $X$, let $K$ be the associated finite simplicial or cubical complex with $N_k$ $k$-cells, and let $f_X : K \to \mathbb{R}$ be the corresponding monotone filtration function. Fix $T_{\min} < T_{\max}$ and an evenly spaced grid $t_{\text{seq}} = \{t_1, \ldots, t_v\}$ on $[T_{\min}, T_{\max}]$ as in Algorithm 1. Define the processed set $K(T_{\max}) := \{\sigma \in K : f_X(\sigma) \le T_{\max}\}$ and let $T(K(T_{\max}), v)$ denote the runtime of Algorithm 1 on this input. Write $Z_\sigma = \mathbb{1}\{f_X(\sigma) \le T_{\max}\}$ and, for a $k$-cell, $p_k = \mathbb{P}(Z_{\sigma_k} = 1)$. Then there exist constants $a, b > 0$ such that:*

(i) $T(K(T_{\max}), v) \;\le\; a\,|K(T_{\max})| + b\,v.$

(ii) $\mathbb{E}\,T(K(T_{\max}), v) \;\le\; a\sum_k p_k N_k + b\,v.$

Proposition 3.1 provides a unified characterization of the ECC computation cost by establishing deterministic and expected bounds for the runtime of Algorithm 1. In the proposition, $N$ can be viewed as shorthand for $|K(T_{\max})|$, i.e., the number of simplices whose filtration values lie below $T_{\max}$. Proposition 3.1 thus indicates a substantial improvement over the $O(N^2 \log^3 N \log\log N)$ time of PH computation (Chen & Kerber, 2013; Boissonnat et al., 2018). This theoretical guarantee complements the empirical findings reported in Section 6, which demonstrate a substantial speedup over both DECT and PH-based approaches. The proof of Proposition 3.1 is provided in Appendix F.1.

### 3.2  Computation of Layer Output: $\mathcal{E} \to \mathcal{O}_\theta$

Upon obtaining the vectorized ECC $\mathcal{E}$, we apply a differentiable parametrized map $g_\theta$ to project it to a learnable task-optimal representation. Given $\mathcal{E} \in \mathbb{R}^v$ and an output dimension $m$, the map $g_\theta : \mathbb{R}^v \to \mathbb{R}^m$ takes $\mathcal{E}$ as input and outputs $\mathcal{O}_\theta \in \mathbb{R}^m$. The structure of $g_\theta$ is unrestricted, provided that it is differentiable with respect to $\theta$ and the layer input. When post-processing is unnecessary, an identity mapping suffices.

## 4  Toward Efficient Backpropagation

To develop a general topological layer suitable for deep learning, establishing differentiability is a fundamental prerequisite. Thus, we first present a differentiability result for the ECC. Using chain rule, we decompose the derivative into two essential components: (i) the derivative of the filtration value with respect to the input $X$, and (ii) the derivative of the indicator function with respect to the filtration value, as shown below.

$$\frac{\partial \chi(K(t))}{\partial X} = \sum_{k=0}^{\infty} (-1)^k \sum_{\sigma \in K^k} \underbrace{\frac{\partial f(\sigma)}{\partial X}}_{(i)} \underbrace{\frac{\partial \mathbb{1}(f(\sigma) \le t)}{\partial f(\sigma)}}_{(ii)}.$$

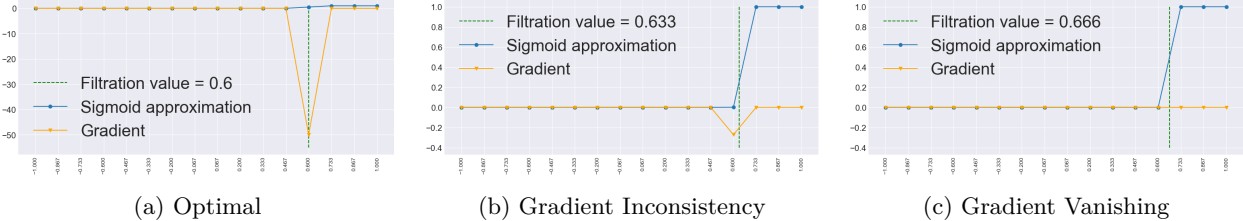

(a) Optimal $\qquad$ (b) Gradient Inconsistency $\qquad$ (c) Gradient Vanishing

Figure 2: Illustration of gradient inconsistency under sigmoid approximation with $\lambda = 200$, evaluated at 16 evenly spaced points over $[-1, 1]$. In (a), the approximation performs well when the filtration value $f(\sigma)$ aligns with a grid point. In (b), slightly shifting $f(\sigma)$ causes a sharp drop in the gradient magnitude, and in (c), a further shift leads to gradient vanishing. This highlights the sensitivity of gradients to the filtration value's position relative to the grid.

In the above formulation, (i) depends on the choice of filtration. While our framework accommodates any differentiable filtration, we restrict our attention to three filtrations that are widely used in practice: Vietoris–Rips, Alpha, and sub/superlevel set filtrations on filtered cubical complexes. Detailed derivation of (i) for the aforementioned filtrations is provided in Appendix D.

## 4.1 Limitations of Sigmoid-Based Approximations

The key barrier in achieving differentiability arises from the discontinuous nature of the indicator function $\mathbb{1}(f(\sigma) \leq t)$ in (ii). To address this, Röell & Rieck (2024) replaced the indicator function with a smooth sigmoid function $S(\lambda(t - f(\sigma)))$, where the hyperparameter $\lambda$ controls the precision of the approximation. However, such smoothing-based approximation increases computational costs, as it requires evaluating the sigmoid function at all grid points $t_i \in tseq$ during each iteration. Consequently, the time complexity of the forward pass escalates to $O(vN)$, compared to the $O(N + v)$ complexity of Algorithm 1 (Proposition 3.1). Furthermore, it introduces challenges during backpropagation, as gradients are computed over a discretized grid rather than a continuous domain. When a filtration value $f(\sigma)$ falls between grid points, the gradient with respect to $f(\sigma)$ is evaluated at neighboring grid locations rather than its exact position. As a result, the magnitude of the total gradients backpropagated through the layer varies with the proximity of $f(\sigma)$ to its adjacent grid points, leading to what we refer to as *gradient inconsistency* (see Figure 2).

In particular, we show that the sigmoid approximation is prone to vanishing gradients when the number of grid points $v$ is too small, resulting in wide grid spacing, or when $\lambda$ is too large (see Figure 2-(c)). To formalize this, consider approximating the gradient of $\mathbb{1}(f(\sigma) \leq t)$ with respect to $f(\sigma)$ on an equally spaced grid $tseq = \{t_1, \ldots, t_v\}$, with spacing $\Delta t \coloneqq \frac{t_{i+1} - t_i}{2}$. Let $S'^{tseq}_{\lambda, f(\sigma)} \in \mathbb{R}^v$ denote the gradient vector of the sigmoid approximation $S(\lambda(t - f(\sigma)))$ computed at $t_1, \ldots, t_v$, i.e., $S'^{tseq}_{\lambda, f(\sigma)} = \frac{\partial S(\lambda(t - f(\sigma)))}{\partial f(\sigma)}\Big|_{t = t_1, \ldots, t_v}$. The following proposition demonstrates that the local gradient of the sigmoid approximation can become arbitrarily small, irrespective of the true underlying gradient.

**Proposition 4.1.** *For fixed $\Delta t$, suppose that $\lambda \to \infty$ to obtain a tight sigmoid approximation of the indicator function. Alternatively, for fixed $\lambda$, suppose that $\Delta t \to \infty$ so that the grid spacing becomes very wide. Then,*

$$\inf_{f(\sigma) \in [t_1 - \Delta t, t_v + \Delta t)} \left\| S'^{tseq}_{\lambda, f(\sigma)} \right\|_\infty \to 0.$$

This highlights a key limitation of the sigmoid approximation, as vanishing gradients hinder effective backpropagation in deep learning due to successive multiplications of local gradients in ECC. The proof is provided in Appendix F.2.

## 4.2 Backpropagation via Distributional Derivatives

We now introduce a novel backpropagation strategy that enhances computational efficiency by facilitating the implementation of Algorithm 1, while simultaneously resolving the vanishing gradient issue. Rather than

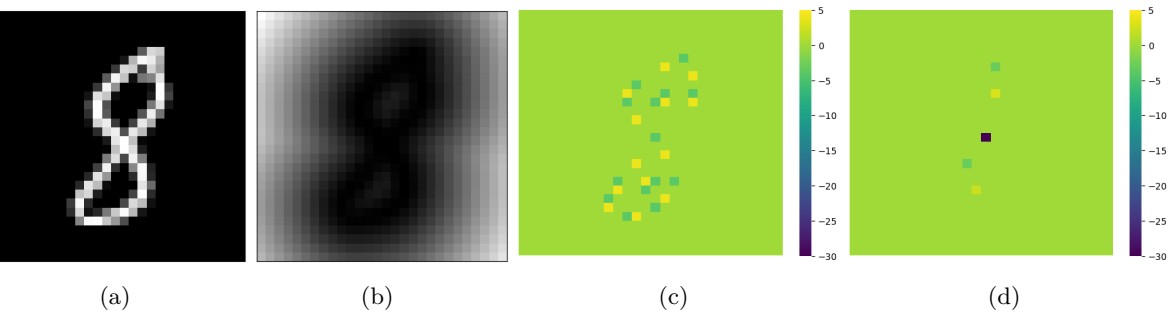

$$(a) \qquad\qquad (b) \qquad\qquad (c) \qquad\qquad (d)$$

Figure 3: (a) is a sample data from the MNIST dataset. (b) represents the DTM transformation of (a) using $m_0 = 0.05$ on a $28 \times 28$ unit grid. (c) and (d) visualize the respective gradients of the ECC and the persistence landscape with respect to (b).

approximating the indicator function with a smooth surrogate, we directly target its gradient by adopting distributional derivatives: $\frac{\partial \mathbb{1}(f(\sigma) \leq t)}{\partial f(\sigma)} = -\delta(t - f(\sigma))$, where $\delta(\cdot)$ denotes the Dirac delta function defined as $\delta(x) = \lim_{\beta \to 0} \frac{1}{|\beta|\sqrt{\pi}} e^{-(x/\beta)^2}$. Since the height of this single impulse at $t = f(\sigma)$ is infinite, we proceed with the approximation $\max_x \frac{1}{|\beta|\sqrt{\pi}} e^{-(x/\beta)^2} = \frac{1}{\beta\sqrt{\pi}}$, where $\beta > 0$ is a hyperparameter that determines the height of the spike. Namely, the estimated gradient evaluates to $-1/(\beta\sqrt{\pi})$ at the impulse point $t = f(\sigma)$, and zero elsewhere. Moreover, to address cases where the impulse location $t = f(\sigma)$ does not align with the predefined grid $tseq$, we shift the impulse to the nearest grid points to ensure proper alignment. Suppose that $t_{i-1} \leq f(\sigma) < t_i$ for $t_{i-1}, t_i \in tseq$, and let $2\Delta t := t_i - t_{i-1}$ be the distance between neighboring grid points. Then, we split the gradient $-1/(\beta\sqrt{\pi})$ by assigning larger weights to the grid point closer to $f(\sigma)$. Specifically, we backpropagate

$$-\left(\frac{t_i - f(\sigma)}{2\Delta t}\right)\frac{1}{\beta\sqrt{\pi}} \quad \text{to grid point } t_{i-1}, \qquad -\left(1 - \frac{t_i - f(\sigma)}{2\Delta t}\right)\frac{1}{\beta\sqrt{\pi}} \quad \text{to grid point } t_i.$$

Under this formulation, a consistent amount of total gradient invariably flows through the discretized grid points, ensuring that a stable and meaningful signal is propagated to the preceding layer (see Figure 3 for a visualization of the gradients). The following proposition demonstrates that our method avoids the gradient vanishing issues inherent to the sigmoid approximation.

**Proposition 4.2.** *Let $\delta_{\beta,f(\sigma)}^{tseq} \in \mathbb{R}^v$ be our gradient approximation of $\frac{\partial \mathbb{1}(f(\sigma) \leq t)}{\partial f(\sigma)}$ computed at $t_1, \ldots, t_v$, such that $\delta_{\beta,f(\sigma)}^{tseq}$'s $i-1$-th and $i$-th elements are respectively $-\left(\frac{t_i - f(\sigma)}{2\Delta t}\right)\frac{1}{\beta\sqrt{\pi}}$ and $-\left(1 - \frac{t_i - f(\sigma)}{2\Delta t}\right)\frac{1}{\beta\sqrt{\pi}}$ if $f(\sigma) \in [t_{i-1}, t_i)$, and other elements are 0. Then, the infimum of the $L_\infty$ norm of $\delta_{\beta,f(\sigma)}^{tseq}$ is given as*

$$\inf_{f(\sigma) \in [t_1 - \Delta t, t_v + \Delta t)} \left\| \delta_{\beta,f(\sigma)}^{tseq} \right\|_\infty = \frac{1}{2\beta\sqrt{\pi}}.$$

The proof is provided in Appendix F.2. In addition to mitigating vanishing gradients, our proposed approach achieves significantly lower approximation error in estimating true gradient values. Suppose the algorithm to approximate the gradients of the indicator function outputs approximations $\frac{\partial \mathbb{1}(f(\sigma) \leq t)}{\partial f(\sigma)}\big|_{t=t_1, \ldots, t_v}$ as $g_1, \ldots, g_v$. We treat that the gradient $\frac{\partial \mathbb{1}(f(\sigma) \leq t)}{\partial f(\sigma)}$ at $t$ is approximated as $g_1$ on $t \in [t_1 - \Delta t, t_1 + \Delta t)$, $g_2$ on $t \in [t_2 - \Delta t, t_2 + \Delta t)$, and $g_v$ on $t \in [t_v - \Delta t, t_v + \Delta t)$, where $g_i$ can depend on $f(\sigma)$. If $g$ is a good approximation of $\frac{\partial \mathbb{1}(f(\sigma) \leq t)}{\partial f(\sigma)}$, then $\int g(t)df(\sigma) \approx \int \frac{\partial \mathbb{1}(f(\sigma) \leq t)}{\partial f(\sigma)}df(\sigma)$ for each $t \in [t_1 - \Delta t, t_v + \Delta t)$, and $\int g(t)dt \approx \int \frac{\partial \mathbb{1}(f(\sigma) \leq t)}{\partial f(\sigma)}dt$ for each $f(\sigma) \in [t_1 - \Delta t, t_v + \Delta t)$. We will analyze the approximations of the gradients based on these criteria.

For given $f(\sigma) \in [t_1 - \Delta t, t_v + \Delta t)$, let the sigmoid gradient approximation be $\hat{S}'_{\lambda,f(\sigma)} : [t_1 - \Delta t, t_v + \Delta t)$ as

$$\hat{S}'_{\lambda,f(\sigma)}(t) = \left(S'^{tseq}_{\lambda,f(\sigma)}\right)_i = -\lambda \cdot S(\lambda(t_i - f(\sigma)))\left[1 - S(\lambda(t_i - f(\sigma)))\right],$$

for $t \in [t_i - \Delta t, t_i + \Delta t)$. Similarly, let our gradient approximation be $\hat{\delta}_{\beta, f(\sigma)} : [t_1 - \Delta t, t_v + \Delta t)$ as

$$\hat{\delta}_{\beta, f(\sigma)}(t) = \left( \delta_{\beta, f(\sigma)}^{tseq} \right)_i = \begin{cases} -\left( \frac{t_{i+1} - f(\sigma)}{2\Delta t} \right) \frac{1}{\beta\sqrt{\pi}}, & \text{if } f(\sigma) \in [t_i, t_{i+1}), \\ -\left( 1 - \frac{t_i - f(\sigma)}{2\Delta t} \right) \frac{1}{\beta\sqrt{\pi}}, & \text{if } f(\sigma) \in [t_{i-1}, t_i), \\ 0, & \text{otherwise,} \end{cases}$$

for $t \in [t_i - \Delta t, t_i + \Delta t)$. The following propositions characterize how closely the two gradient methods approximate the true underlying gradient.

**Proposition 4.3.** *For all* $t \in [t_1 - \Delta t, t_v + \Delta t)$,

$$\left| \int_{t_1 - \Delta t}^{t_v + \Delta t} \hat{S}'_{\lambda, f(\sigma)}(t) df(\sigma) - \int_{t_1 - \Delta t}^{t_v + \Delta t} \frac{\partial \mathbb{1}(f(\sigma) \leq t)}{\partial f(\sigma)} df(\sigma) \right| \geq 2S(-\lambda v \Delta t). \tag{4}$$

*And if* $f(\sigma) = t_i - \Delta t$ *for some* $t_i \in tseq$, *then*

$$\left| \int_{t_1 - \Delta t}^{t_v + \Delta t} \hat{S}'_{\lambda, f(\sigma)}(t) dt - \int_{t_1 - \Delta t}^{t_v + \Delta t} \frac{\partial \mathbb{1}(f(\sigma) \leq t)}{\partial f(\sigma)} dt \right| \geq |1 - 2\lambda v \Delta t \exp(-\lambda \Delta t)|. \tag{5}$$

**Proposition 4.4.** *For all* $t \in [t_1 - \Delta t, t_v + \Delta t)$,

$$\int_{t_1 - \Delta t}^{t_v + \Delta t} \hat{\delta}_{\beta, f(\sigma)}(t) df(\sigma) - \int_{t_1 - \Delta t}^{t_v + \Delta t} \frac{\partial \mathbb{1}(f(\sigma) \leq t)}{\partial f(\sigma)} df(\sigma) = -\frac{2\Delta t}{\beta\sqrt{\pi}} + 1,$$

*and for all* $f(\sigma) \in [t_1 - \Delta t, t_v + \Delta t)$,

$$\int_{t_1 - \Delta t}^{t_v + \Delta t} \hat{\delta}_{\beta, f(\sigma)}(t) dt - \int_{t_1 - \Delta t}^{t_v + \Delta t} \frac{\partial \mathbb{1}(f(\sigma) \leq t)}{\partial f(\sigma)} dt = -\frac{2\Delta t}{\beta\sqrt{\pi}} + 1.$$

Suppose the grid is fixed, so $\Delta t$ and $v$ is fixed. Then for equation 4 to go to 0, $\lambda \to \infty$ should hold. However, as $\lambda \to \infty$, the lower bound of equation 5 converges to 1, which means that the integral of the sigmoid approximation $\int_{t_1 - \Delta t}^{t_v + \Delta t} \hat{S}'_{\lambda, f(\sigma)}(t) dt$ becomes inconsistent. This is already expected from the vanishing gradient behavior. However, Proposition 4.4 suggests that when $\beta$ is appropriately chosen as $\beta = \frac{2\Delta t}{\sqrt{\pi}}$, the gradient approximation becomes consistent for the integral with respect to both $f(\sigma)$ and $t$. The proofs for Proposition 4.3 and 4.4 are provided in Appendix F.2.

# 5 Stability Analysis

An essential benefit of using a topological layer is its robustness against noise. Extending the results of Dłotko & Gurnari (2023), we can establish a stability property for the layer output with respect to changes in the input. For notation, let $X, X'$ be two distinct inputs, and $f_X, f_{X'}$ be corresponding filtration functions on fixed simplicial complexes $K$, $K'$, respectively. Let $\mathcal{D}_k(X), \mathcal{D}_k(X')$ be corresponding $k$-dimensional persistence diagrams, and let $\mathcal{C}_X, \mathcal{C}_{X'} : \mathbb{R} \to \mathbb{R}$ be corresponding ECC functions. See Appendix C.1 for the definition of Wasserstein distance.

We first see the relation between the final layer output and ECC functions. The precise proof is provided in Appendix F.3.

**Proposition 5.1.** *Let* $t_1^* < t_2^* < \cdots < t_w^*$ *be unique values of all births and deaths in* $\{\mathcal{D}_k(X), \mathcal{D}_k(X') : k \geq 0\}$, *and let* $tseq = \{t_1, \ldots, t_v\}$. *Suppose there exists* $\Delta t > 0$ *satisfying* $\Delta t \leq t_{i+1} - t_i$ *and* $\Delta t \leq 2(t_{i+1}^* - t_i^*)$. *Let* $g_\theta$ *be* $L$-*Lipschitz with respect to* $\|\cdot\|_1$-*norm, i.e.,* $\|g_\theta(x) - g_\theta(y)\|_1 \leq L \|x - y\|_1$. *Then*

$$\|\mathcal{O}_\theta(X) - \mathcal{O}_\theta(X')\|_1 \leq \frac{2L}{\Delta t} \|\mathcal{C}_X - \mathcal{C}_{X'}\|_1.$$

Once $\Delta t$ is specified from the data and the chosen grid, Proposition 5.1 provides a generally applicable bound on the layer outputs in terms of the ECC functions. Hence, the key requirement is to establish the stability of ECC functions. We begin by presenting the most general stability result, stated with respect to the 1-Wasserstein distance between the persistence diagrams of the inputs, following directly from Dłotko & Gurnari (2023, Proposition 3.2).

$$\|\mathcal{C}_X - \mathcal{C}_{X'}\|_1 \le 2 \sum_{k=0}^{\infty} W_1(\mathcal{D}_k(X), \mathcal{D}_k(X')). \tag{6}$$

The behavior of the 1-Wasserstein distance $W_1(\mathcal{D}_k(X), \mathcal{D}_k(X'))$ is in general complicated and difficult to analyze. It is possible to further upper bound this by the difference of the filtration functions $f_X$ and $f_{X'}$. The difference is represented as $L_\infty$ distance below, but there is a more general version of Theorem 5.2 as well (see Appendix E).

**Theorem 5.2.** *Suppose $K = K'$ and is a finite simplicial or cubical complex. Then, there exists a constant $C_K$ only depending on $K$ such that*

$$\|\mathcal{C}_X - \mathcal{C}_{X'}\|_1 \le C_K \|f_X - f_{X'}\|_\infty.$$

Theorem 5.2 provides a stability result whose relation to the difference of the input is clear, and also applicable to general filtration functions. Since we use DTM functions in Section 6, we present a specific result for DTM. The DTM function is a distance-like function that is robust to noise and outliers, whose precise definition is provided in Appendix C.1. The proof of Theorem 5.2 is provided in Appendix F.3.

**Corollary 5.3.** *Suppose $K$ is a finite cubical complex, and $f_X$, $f_{X'}$ are restrictions of DTM functions $d_{P_X, m_0}, d_{P_{X'}, m_0}$ to $K$, where $P_X, P_{X'}$ are empirical distributions on $X$ and $X'$, respectively (for detailed meaning, see Appendix F.3). Then*

$$\|\mathcal{C}_X - \mathcal{C}_{X'}\|_1 \le \frac{C_K}{\sqrt{m_0}} W_2(P_X, P_{X'}).$$

Corollary 5.3 establishes a stability bound at the level of empirical distributions, implying that the corresponding ECCs remain close whenever the empirical distributions are similar. Due to the inherent reliance of Euler characteristics on even small generators, we note that the above stability results in terms of the Wasserstein distance are less strict than those of PH-based descriptors bounded by the Bottleneck distance in Kim et al. (2020). Thus, ECC-based descriptors compromise stability in order to attain computational efficiency over PH-based descriptors. The proof of Corollary 5.3 is provided in Appendix F.3.

In practical settings, filtrations are frequently corrupted by random noise (e.g., sensor noise, missingness, or stochastic augmentations). To align the analysis with these scenarios, we next establish stochastic stability bounds that quantify both the expected fluctuation and high-probability concentration of the difference in ECC functions, when the filtration is perturbed by bounded noise. In what follows, we let $X$ and $\tilde{X}$ denote the original and (stochastically) perturbed inputs, respectively.

**Theorem 5.4.** *Let $K$ be a fixed finite simplicial or cubical complex and write $N = |K|$. Suppose $f_X : K \to \mathbb{R}$ is a filtration for input $X$ such that, for all $\sigma \in K$ and some $\epsilon > 0$, $\max_{\tau \le \sigma}\{f(\sigma) - f(\tau)\} \le \epsilon$, where $\tau \le \sigma$ denotes that $\tau$ is a face of $\sigma$. Now, consider a random perturbation $f_{\tilde{X}} = f_X + \xi$, where $\{\xi(\sigma)\}_{\sigma \in K}$ are independent, mean-zero random variables with bounded support on $[-\epsilon/2, \epsilon/2]$. Let $\mathcal{C}_X$ and $\mathcal{C}_{\tilde{X}}$ denote the corresponding ECCs. Then there is a constant $C_K > 0$ depending only on $K$ such that:*

1. *$\|\mathcal{C}_X - \mathcal{C}_{\tilde{X}}\|_1 \le C_K \, \epsilon/2 \quad a.s.,$*

2. *$\mathbb{E}\big[\|\mathcal{C}_X - \mathcal{C}_{\tilde{X}}\|_1\big] \le C_K \, \epsilon/2,$*

3. *$\mathbb{P}(\|\mathcal{C}_X - \mathcal{C}_{\tilde{X}}\|_1 > \varepsilon) \le 2N \exp\Big(-\frac{\varepsilon^2}{2C_K^2 \epsilon^2}\Big),$*

*for all $\varepsilon > 0$.*

Theorem 5.4 provides almost sure, expected, and high-probability bounds for small stochastic perturbations whose magnitude is controlled by the minimum filtration value difference between all simplices and their faces. Although the stochastic stability results are applicable to a broad class of bounded perturbations commonly encountered in practice, they rely on a strict separation between the filtration values of each simplex and its faces. Consequently, these results do not directly extend to cubical complexes constructed using standard $V$- or $T$-constructions, where cells and their faces share identical filtration values. The proof is provided in Appendix F.3. Next, combining Theorem 5.4 with Proposition 5.1 yields an immediate output-level stability result. This quantifies how random perturbations in the filtration affect the expected and high-probability behavior of $\mathcal{O}_\theta(X)$, as formally stated in the following corollary.

**Corollary 5.5.** *Let $g_\theta$ be L-Lipschitz with respect to $\|\cdot\|_1$ and let $\Delta t$ satisfy the discretization conditions in Proposition 5.1. Then,*

$$\mathbb{E}\big[\|\mathcal{O}_\theta(X) - \mathcal{O}_\theta(\tilde{X})\|_1\big] \ \leq \ \frac{2L}{\Delta t}\, C_K\, \epsilon/2$$

*and for every $\varepsilon > 0$,*

$$\mathbb{P}\big(\|\mathcal{O}_\theta(X) - \mathcal{O}_\theta(\tilde{X})\|_1 > \varepsilon\big) \ \leq \ 2N \exp\Big(-\frac{\varepsilon^2\,\Delta t^2}{8\, L^2\, C_K^2\, \epsilon^2}\Big).$$

The proof is provided in Appendix F.3.

# 6 Experiments

To showcase the versatility and effectiveness of our layer, we conduct a series of experiments. First, we evaluate the computational efficiency of our approach by measuring runtime metrics across different datasets. Next, we implement a topological autoencoder on point cloud data to illustrate how our layer can impose topological constraints on the latent space. Finally, we perform two classification tasks on image, point cloud, and voxel-based datasets. The first classification task demonstrates that our layer can help mitigate information loss under a controlled, systematic data contamination process. The subsequent experiment highlights the advantages of our layer in high-dimensional settings, where PH-based methods would otherwise incur substantial computational cost. Here, we provide only a partial summary of the experimental results. For comprehensive results, including detailed hyperparameter settings, see Appendix B.

## 6.1 Computational Efficiency

In this section, we analyze the empirical time complexity of our method in comparison to PH and sigmoid-approximated ECC, hereafter referred to as DECC. The time complexity of each topological descriptor is assessed by measuring the runtime of a complete iteration through the training dataset, averaged over 10 repetitions. To investigate how each descriptor scales with increasing dimensionality, we consider three datasets with varying input sizes: MNIST (size $28 \times 28$), Fracture3D (size $28 \times 28 \times 28$), and synthetic data (size $112 \times 112$) generated by randomly sampling each pixel from a uniform distribution. PH is computed using the `GUDHI` library (The GUDHI Project, 2021), while ECC and DECC are implemented according to Algorithm 1; for DECC, steps 9-10 are replaced with applying the sigmoid function $S(\lambda(t_i - f(\sigma)))$ at each grid point $t_i$, and step 11 is omitted.

**Result.** The experimental results in Table 1 indicate that ECC dramatically improves both runtime and scalability relative to PH, achieving an $8{\sim}20$ times speedup. Given the additional computational overhead typically required to convert PH into representations amenable to subsequent machine learning models, our approach provides a substantial advantage over all PH-based methods in terms of efficiency, both theoretically and empirically. Moreover, as discussed in Section 4.1, empirical results evidence that removing the sigmoid approximation yields a notable improvement in runtime, further highlighting the practicality of our design.

## 6.2 Topological Autoencoder

The idea of imposing topological constraints on the latent space was first explored by Hofer et al. (2019); Moor et al. (2020). Whereas existing works rely on a topology-based loss term to regularize the latent space,

Table 1: Average runtime per iteration (in seconds).

| Data (Dimension) (Sample Size) | ECC | DECC | PH |
|---|---|---|---|
| MNIST ($28 \times 28$) ($60,000$) | 2.06 sec | 14.02 sec | 17.20 sec |
| Synthetic ($112 \times 112$) ($1,000$) | 0.43 sec | 4.00 sec | 6.81 sec |
| Fracture3D ($28 \times 28 \times 28$) ($1,027$) | 1.42 sec | 12.59 sec | 28.71 sec |

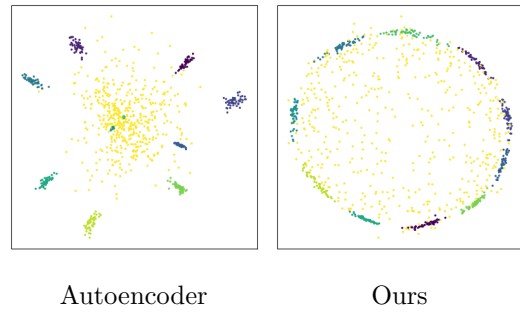

Autoencoder        Ours

Figure 4: Latent representation of Spheres data.

our formulation allows for the utilization of standard loss functions, such as Mean Squared Error (MSE) or Mean Absolute Error (MAE), to achieve a similar goal. Motivated by the stability results with respect to the $L_1$ distance in Section 5, we impose a topological constraint using the MAE loss between the ECC of the input data and the ECC of its latent representation. The respective ECCs are computed using Vietoris-Rips filtration, with maximum dimension set to one. For this experiment, we use the synthetic Spheres dataset from Moor et al. (2020), which comprises ten 100-spheres of radius $r = 5$ enclosed by one larger 100-sphere of radius $r = 25$, all embedded in 101-dimension. The ten smaller spheres are shifted in random directions according to Gaussian noise, and we set the latent space dimension to two.

**Result.** We discover that our approach effectively preserves the shape of the encompassing sphere (yellow points in Figure 4), while the vanilla autoencoder fails to retain this structure. Moreover, our method constrains the smaller spheres to remain on the boundary of the encompassing sphere, whereas in the vanilla autoencoder, numerous smaller circles lie far beyond the boundaries of the encompassing circle. However, with this simplistic architecture, the model's capacity to comprehensively articulate the nested relationship inherent in the data was somewhat restricted. While our method demonstrates capability of regularizing the latent space, we do not claim superiority over alternative approaches. Rather, we present it as a motivating example of how topological characterization in the latent space can be promoted via simple standard loss functions. Additional quantitative evaluations of the dimensionality reduction quality are provided in Appendix B.2.

### 6.3 Data Contamination

Next, we perform classification under data contamination using the MNIST image dataset and the ORBIT5K point cloud dataset. Following Kim et al. (2020), with probability $0.0, 0.05, \ldots, 0.2$, we first corrupt the data by randomly omitting pixels or points, and subsequently add uniformly-distributed noise at random. Although this procedure is likely too simplistic to fully reflect real-world data contamination scenarios, it provides a controlled setting for a preliminary study of how our model responds to systematic corruption and noise. Thus, our primary interest in this section is to demonstrate that, under this particular contamination setup, our layer (i) consistently outperforms the vanilla baseline and exhibits increased robustness to noise, and (ii) attains performance comparable to other topological models while substantially reducing computational cost. To impartially illustrate the advantages of our layer, we deliberately retain a simplified experimental setting, featuring small training sets (1000 samples for both MNIST and ORBIT5K) and a simple vanilla baseline model. 30% of the training data is used as a validation set, and model performance is evaluated on the full test data. We report the average test accuracy and runtime (per epoch) over 15 runs, where the runtime within each run is computed as the total training time divided by the number of training epochs.

**Experimental Setup.** The vanilla baseline model consists of two CNN layers followed by two fully connected layers. We compare the performance of our proposed layer to the vanilla baseline model, and two other PH-based topological layers applicable to image datasets: PersLay (Carrière et al., 2020) and PLLay (Kim et al., 2020). To directly compare the differentiability strategy underlying DECT with our proposed approach, we additionally implement DECC, a sigmoid-approximated variant of ECLayr, as a surrogate for

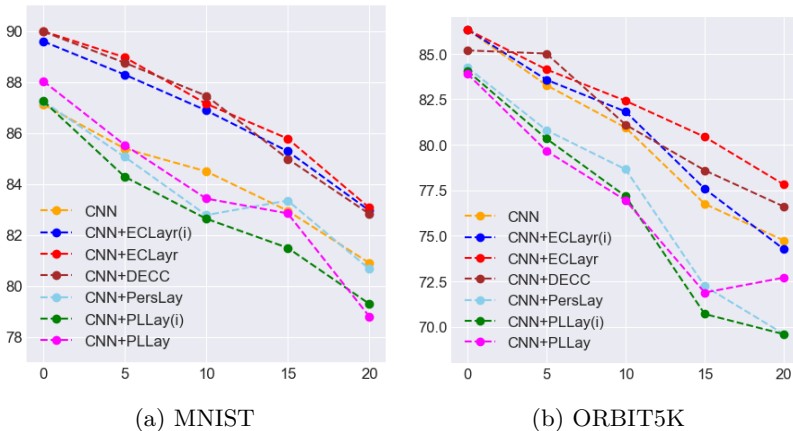

(a) MNIST   (b) ORBIT5K

Figure 5: Test accuracy on MNIST and ORBIT5K datasets. Performance in the presence of different levels of noise.

Table 2: Average runtime per epoch over 1000 MNIST data (in seconds).

| Model | Runtime |
|---|---|
| CNN | 0.112 sec |
| CNN + ECLayr(i) | 0.165 sec |
| CNN + ECLayr | 0.279 sec |
| CNN + DECC | 0.884 sec |
| CNN + PersLay | 9.311 sec |
| CNN + PLLay(i) | 8.799 sec |
| CNN + PLLay | 17.630 sec |

DECT (Röell & Rieck, 2024). While DECC does not incorporate the full ECT over multiple directions and is not intended to reproduce DECT's geometry-specific design tailored to graphs and meshes, it serves as a natural proxy for isolating the effect of the differentiability mechanism itself. Each topological layer is placed in parallel at the beginning of the network (referred to as CNN + ECLayr(i), PersLay, PLLay(i)). For topological layers that allow backpropagation, we add an additional layer after the last convolutional layer (referred to as CNN + ECLayr, DECC, PLLay). A visualization of the model architecture is provided in Figure 7-(a) of Appendix B.

**Result.** As shown in Figure 5, ECLayr generally outperforms other baseline models for both datasets. Interestingly, ECLayr outperforms PH-based models in this setting, despite PH being theoretically more informative than ECC. We conjecture that ECC benefits from its relative simplicity, particularly when applied to datasets with straightforward topological structures, as discussed in Section 2.2, or in small-sample settings, as in our experiment. While the expressivity of PH can be advantageous for modeling intricate underlying distributions, it may be unnecessary, or even detrimental due to overfitting, when the target topology is simple or when data is limited. In these cases, ECC may be easier to learn than PH, both from statistical and optimization standpoints. Consequently, the simplicity of ECC makes our layer particularly well-suited for scenarios in which the essential underlying topology can be effectively captured by ECC. Moreover, our method scales approximately $50 \sim 60$ times faster than PH-based approaches (see Table 2), underscoring its substantial computational efficiency. It also achieves comparable, and in some cases slightly superior, performance to DECC, while providing a threefold speedup by eliminating the need for sigmoid approximation. Figure 5 further shows that, under the designed noise protocol, ECLayr appears robust up to approximately $10 \sim 15\%$ contamination relative to the vanilla baseline for MNIST. For ORBIT5K, this relative robustness is retained even at 20% contamination. See Table 5 and 7 in Appendix B for detailed experiment results.

**Hyperparameter Influence.** To evaluate the influence of each hyperparameter on performance, we conduct an ablation study on MNIST and ORBIT5K by varying a single hyperparameter while keeping all others fixed. The test accuracy of CNN + ECLayr across different hyperparameter settings is presented in Table 6 and 8 of Appendix B.

### 6.4  High-Dimensional Data

A key advantage of ECC lies in its computational efficiency, which enables the incorporation of topological insights in complex, high-dimensional settings where PH-based methods often become computationally prohibitive. To evaluate the scalability of ECLayr in such contexts, we conduct experiments on several 3D biomedical voxel datasets from MedMNIST3D (Yang et al., 2023).

Table 3: Test accuracy and runtime per epoch (in seconds) on MedMNIST3D datasets.

| Models | Fracture3D | | Organ3D | | Synapse3D | | Vessel3D | |
| --- | --- | --- | --- | --- | --- | --- | --- | --- |
| | Test Acc. | Time | Test Acc. | Time | Test Acc. | Time | Test Acc. | Time |
| ResNet | 53.42 ($\pm4.61$) | 5.183 sec | 88.20 ($\pm4.54$) | 4.980 sec | 76.31 ($\pm3.57$) | 6.107 sec | 90.13 ($\pm2.36$) | 6.561 sec |
| ResNet + ECLayr(i) | 54.13 ($\pm2.92$) | 7.021 sec | 87.57 ($\pm2.22$) | 12.023 sec | 77.10 ($\pm3.14$) | 14.912 sec | 91.41 ($\pm1.19$) | 8.889 sec |
| ResNet + ECLayr | **56.00** ($\pm3.58$) | 14.743 sec | **89.30** ($\pm1.99$) | 29.985 sec | **78.18** ($\pm2.22$) | 27.156 sec | **92.36** ($\pm1.89$) | 20.439 sec |
| ResNet + DECC | 54.33 ($\pm4.98$) | 111.539 sec | 87.57 ($\pm1.97$) | 180.021 sec | 76.39 ($\pm1.56$) | 198.853 sec | 91.41 ($\pm1.99$) | 220.030 sec |

**Experimental Setup.** Following the practice of Yang et al. (2023), we implement ResNet18 (He et al., 2016) with 3D convolution as the vanilla baseline model. A parallel ECLayr is introduced at the beginning of the network for ResNet + ECLayr(i), and we place an additional topological layer after the first residual block for ResNet + ECLayr, DECC. A visualization of the model architecture is provided in Figure 7-(b) of Appendix B. Each simulation is repeated 10 times, with the average test accuracy and runtime (per epoch) reported in Table 3. The runtime is computed as in Section 6.3. Due to limited computational resources and the 20-fold longer runtime of PH reported in Table 1, PH-based models were too costly for this high-dimensional experiment and were therefore omitted. To further reduce computational cost, we replace the DECC at the beginning of the ResNet+DECC model with ECLayr, since this layer does not require backpropagation.

**Result.** The results in Table 3 reveal that exploiting topological information via ECLayr leads to performance improvements even on complex, high-dimensional data. These findings further suggest that our layer can be seamlessly integrated into larger architectures, such as ResNet, for practical applications beyond datasets with relatively simple structures. Most importantly, ECLayr consistently outperforms DECC with substantially reduced runtime, demonstrating that the proposed backpropagation strategy is both effective and computationally efficient. Notably, ResNet+ECLayr remains $7 \sim 11$ times faster than ResNet+DECC even after replacing the first DECC with ECLayr to reduce computational cost, highlighting the efficiency of our approach. These results position ECLayr as a more practical alternative for integrating topological information into complex, high-dimensional data structures, underscoring its relevance for real-world applications.

# 7 Discussion

ECLayr is a novel topological layer that alleviates computational overhead and enables efficient backpropagation, facilitating seamless integration into diverse deep learning architectures. It applies broadly to data structures with differentiable filtrations. Nonetheless, several key limitations merit attention. First, while ECCs offer computational efficiency, PH-based summaries capture richer, multi-scale topological information. Understanding this trade-off is crucial. Accordingly, ECLayr is best suited for applications where efficiency is prioritized over fine-grained topological detail, or settings that admit relatively simple topological structures. Moreover, as detailed in Section 5, ECCs are topologically weaker invariants than PHs, often resulting in less stable representations. As with other topological layers, further research is needed to support systematic hyperparameter tuning. Also, extending our analysis to other filtrations, such as clique complexes on multigraphs, and applying ECLayr to time-series embeddings (Umeda, 2017; Kim et al., 2018) represents a promising direction for future work, further demonstrating the versatility of our approach.

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

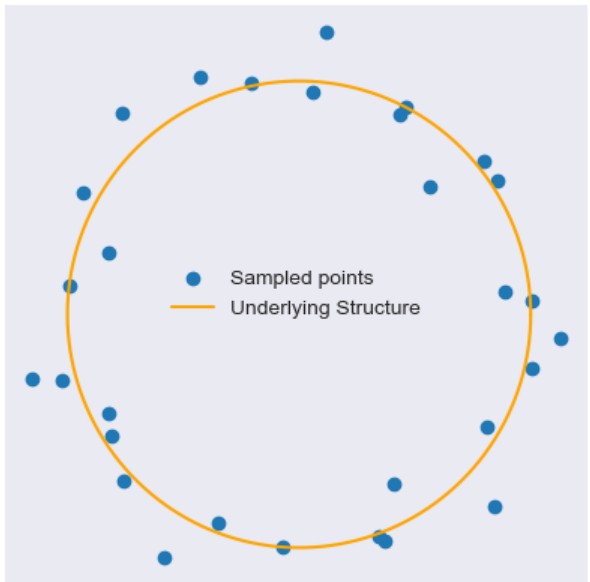

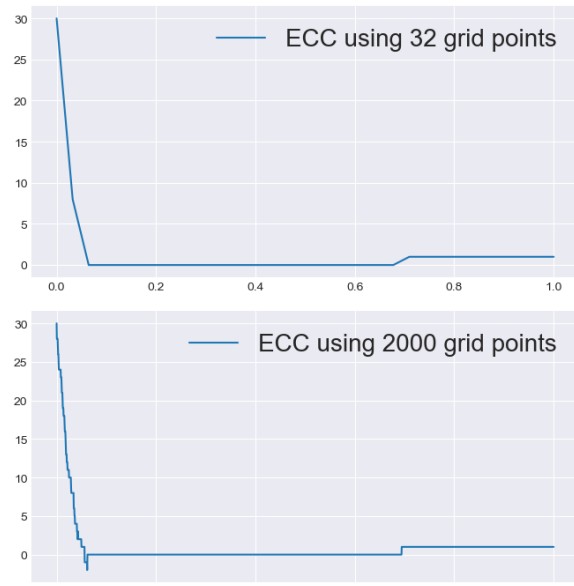

(a) Thirty points uniformly sampled from a unit circle, perturbed by small noise.

(b) ECC of (a) calculated using 32 and 2000 grid points, respectively.

Figure 6: Within the interval $[0.06, 0.7]$, both ECCs capture the Euler characteristic of the underlying loop structure, which is zero. The ECC with 2000 grid points exhibits more noise than the ECC with 32 grid points, as the dense discretization captures even the small (noise) generators that do not represent the global structure of data.

## A    Guideline for Hyperparameter Selection

ECLayr requires the specification of several hyperparameters: filtration, $T_{min}, T_{max}, v$ and $\beta$. In this section, we provide a concise guideline for selecting appropriate hyperparameter values.

**Choice of filtration.** Although a diverse range of filtration options exist, specific filtrations are commonly preferred based on the inherent characteristics of the data and the contextual requirements of the training process. For instance, Vietoris-Rips and Alpha filtrations are extensively utilized for point clouds, whereas sub/superlevel set filtrations on filtered cubical complexes are a natural choice for data with grid structures (e.g., images and voxels). DTM filtration provides robustness against outliers, and thereby preferable in scenarios of data contamination. Although not discussed in detail here, alternative filtration choices are also available. Nevertheless, the core principle is to select a filtration that best leverages the intrinsic characteristics of the data and training process.

**Choice of $[T_{min}, T_{max}]$.** A naive and convenient approach is to assign $T_{min}$ and $T_{max}$ as the minimum and maximum of possible filtration values, respectively. An alternative method is to select $[T_{min}, T_{max}]$ as a tighter interval within the range of possible filtration values, focusing on regions of the filtration that contain meaningful topological and geometrical information. Such an interval can be identified via hyperparameter search, or chosen heuristically by analyzing the ECC computed from data samples. For instance, the ECCs depicted in Figure 6-(b) suggest that the interval $[0.06, 0.7]$ is appropriate for effectively capturing the underlying loop structure in Figure 6-(a).

**Choice of $v$.** Selecting an appropriate grid resolution is crucial, as the vectorized ECC does not account for cycles that are born and perished between adjacent grid points $t_i$ and $t_{i+1}$. However, cycles with very short life spans are often insignificant (noisy) generators. Therefore, excessively dense discretization may not be beneficial, as including numerous small, noisy generators in the learning process can lead to increased variance (see Figure 6-(b)). On the other hand, the discretization should also avoid being overly sparse, as this could compromise the ability to capture essential global features. Although determining the optimal

choice of discretizing bins is a non-trivial task, standard data-driven heuristics, such as cross-validation or a simple grid search, can be employed to select an appropriate value for $v$ that effectively balances the aforementioned trade-offs.

**Choice of $\beta$.** The hyperparameter $\beta$ controls the magnitude of the gradient, where smaller values of $\beta$ result in larger gradient magnitudes. While we suggest $\beta = 2\Delta t/\sqrt{\pi}$ as a reasonable starting point (see Proposition 4.4), a clear theoretical rationale for selecting the optimal value of $\beta$ remains elusive. Analogous to the selection process of $v$, standard data-driven heuristics, such as cross-validation, can be employed to determine an appropriate value of $\beta$ that is empirically optimal.

# B  Experiment Details

The experiments were implemented using `GUDHI` and `Pytorch`. The high-dimensional data experiment in Section 6.4 was executed on an NVIDIA RTX A6000 GPU, and all other experiments were conducted on an Apple M4 Max processor.

## B.1  Computational Efficiency

The runtime metrics are computed on the training sets of MNIST, Fracture3D, and synthetic data. The MNIST training set contains 60000 images of size $28 \times 28$, while the Fracture3D training set consists of 1,027 voxels of size $28 \times 28 \times 28$. We generate 1000 samples of size $112 \times 112$ for the synthetic data, where each pixel is randomly drawn from a uniform distribution. All methods use sublevel set filtration on V-constructed cubical complex. PH is computed using the `GUDHI` package, while Algorithm 1 is used to compute ECC and DECC with $v = 32$ on interval $[0, 1]$; DECC further applies the sigmoid approximation at each iteration.

## B.2  Topological Autoencoder

The encoder and decoder network each consists of three fully connected layers, with input dimension size 101, hidden dimensions size 32, and latent dimension size 2. BatchNorm and ReLu nonlinearity is used after each layer, with the exception of the latent dimension. One ECLayr is applied to the input point cloud while a second ECLayr is applied to the latent representation, thereby computing the MAE loss between the ECC of the input and the ECC of its latent representation. This MAE loss serves as a topological regularizing term, where the regularizer coefficient is 0.1. A Vietoris-Rips filtration is employed to compute the ECC with the maximum dimension restricted to 1. We use 1000 grid points over the interval $[0, 1.05]$. $\beta$, which controls the magnitude of the gradient, is assigned as $\frac{2\Delta t}{\sqrt{\pi}}$ and the post-processing layer $g_\theta$ is set as an identity mapping. Adam optimizer is used for training with batch size 32 and learning rate 0.0001. We run for 200 epochs and adopt early stopping after patience 10. To quantitatively evaluate the quality of the latent representation, we consider the following measures used in Moor et al. (2020):

1. Kullback-Leibler (KL) divergence: measures the divergence between the density estimates of the input and latent space.

2. Root mean square error (RMSE): RMSE between the distance matrix of the original space and the latent space.

3. Mean relative rank error (MRRE): measures the changes in ranks of distances in the original space and the latent space.

4. Trustworthiness: measures how well local neighborhoods, in terms of $k$ nearest-neighbors, are preserved when going from the original space to the latent space.

5. Continuity: measures how well local neighborhoods, in terms of $k$ nearest-neighbors, are preserved when going from the latent space to the original space.

A Gaussian kernel with bandwidth $\sigma > 0$ is used for density estimation when computing the KL divergence, and $\sigma$ is varied for assessment across multiple scales. Table 4 compares the quality of the latent representations learned by the vanilla autoencoder and our autoencoder across multiple evaluation metrics. We observe

Table 4: Quality of latent representations across multiple evaluation metrics. For each metric, the more desirable value is highlighted in bold.

| | KL ($\sigma = 0.01$) | KL ($\sigma = 0.1$) | KL ($\sigma = 1$) | RMSE | MRRE | Trustworthiness | Continuity |
|---|---|---|---|---|---|---|---|
| Autoencoder | 0.388 | 0.822 | 0.019 | 0.666 | 0.285 | **0.666** | 0.785 |
| Ours | **0.320** | **0.519** | **0.015** | **0.562** | **0.273** | **0.666** | **0.807** |

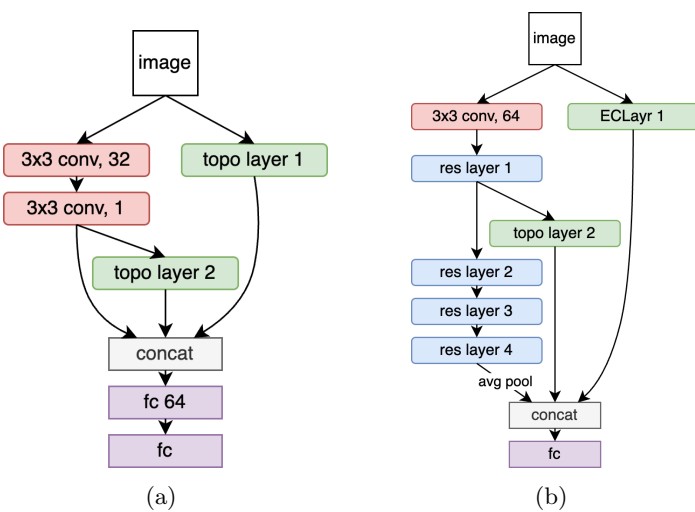

(a)  (b)

Figure 7: (a): Model architecture of CNN + ECLayr, DECC, PLLay for the MNIST and the ORBIT5K experiment. Topo layer 2 is removed for CNN + ECLayr(i), PLLay(i), PersLay. (b): Model architecture of ResNet + ECLayr, DECC for the MedMNIST3D experiment. Topo layer 2 is removed for ResNet + EClayr(i).

that across all metrics, the latent representation of our autoencoder better preserves the structure of the input data.

### B.3 MNIST

**Training setup.** The MNIST dataset contains 60000 training data and 10000 test data of handwritten digits from 0 to 9. We implement a 4 layer vanilla baseline model, consisting of two convolutional layers followed by two fully connected layers, with ReLU nonlinearity between every layer. Both convolutional layers use $3 \times 3$ kernels with stride 1 and padding 1. Each convolutional layer has channel size 32 and 1, respectively. The output of the convolutional network is flattened and passed to the subsequent fully connected layers with hidden dimension of 64. We compare the performance of our proposed layer with the vanilla baseline model, and two other PH-based topological layers applicable to image datasets: PersLay (Carrière et al., 2020) and PLLay (Kim et al., 2020). We additionally implement an ECLayr with sigmoid approximation (referred to as DECC) to facilitate a proxy comparison with DECT (Röell & Rieck, 2024). For all topological layers, we place a parallel layer at the beginning of the network (referred to as CNN + ECLayr(i), CNN + PersLay, and CNN + PLLay(i)). For ECLayr, PLLay, and DECC, which allow backpropagation, we add an additional layer after the last convolutional layer (referred to as CNN + ECLayr, CNN + DECC and CNN + PLLay). A visualization of the model architecture is presented in Figure 7-(a). We train using the Adam optimizer with learning rate 0.001 and batch size 32. Training runs for at most 100 epochs with early stopping after the validation loss plateaus for 5 epochs, and cross-entropy loss is used for classification. We randomly sample 1000 training data with equal proportion for each label, and 30% of the training data is used as a validation set. Model performance is evaluated on the complete test data of 10000 samples. Using a very simple

Table 5: Test accuracy of models trained on 1000 MNIST data with different corruption and noise probability. For each corruption and noise probability, the best accuracy is highlighted in bold.

| Models | Corruption & Noise Probability | | | | |
| --- | --- | --- | --- | --- | --- |
| | 0.00 | 0.05 | 0.10 | 0.15 | 0.20 |
| CNN | 87.12 ($\pm$0.78) | 85.39 ($\pm$1.41) | 84.49 ($\pm$0.80) | 82.95 ($\pm$1.69) | 80.90 ($\pm$2.31) |
| CNN + ECLayr(i) | 89.59 ($\pm$0.38) | 88.29 ($\pm$0.99) | 86.88 ($\pm$1.23) | 85.28 ($\pm$1.06) | 82.96 ($\pm$1.16) |
| CNN + ECLayr | **89.99** ($\pm$0.68) | **88.97** ($\pm$0.65) | 87.15 ($\pm$0.63) | **85.78** ($\pm$0.49) | **83.07** ($\pm$1.73) |
| CNN + DECC | 89.98 ($\pm$0.85) | 88.75 ($\pm$0.90) | **87.46** ($\pm$0.67) | 84.98 ($\pm$0.99) | 82.83 ($\pm$1.05) |
| CNN + PersLay | 87.24 ($\pm$1.15) | 85.06 ($\pm$1.12) | 82.78 ($\pm$2.22) | 83.35 ($\pm$1.16) | 80.66 ($\pm$1.04) |
| CNN + PLLay(i) | 87.25 ($\pm$1.35) | 84.28 ($\pm$2.34) | 82.64 ($\pm$2.25) | 81.49 ($\pm$1.86) | 79.30 ($\pm$1.53) |
| CNN + PLLay | 88.04 ($\pm$1.32) | 85.53 ($\pm$1.04) | 83.42 ($\pm$1.87) | 82.84 ($\pm$1.58) | 78.80 ($\pm$3.38) |

model and a limited training set, we occasionally observed sporadic training failures across all methods, in which optimization fails to converge and the model does not learn; when training succeeds, all models should achieve near 100% training accuracy on this dataset. To remove the influence of random training failures and solely evaluate model performance, we repeat each experiment 15 times *excluding* runs in which training fails, which we define as achieving below 80% training accuracy. Furthermore, as a preliminary study of how our model mitigates information loss under systematic data contamination, we consider a corruption and noise process where the pixels are randomly omitted and subsequently contaminated at random by noise between 0 and 1 with probability $0.05, 0.1, 0.15$, and $0.2$.

**Hyperparameter setting.** For the first topological layer, we align the data on a unit grid and use the DTM filtration in equation 7 with $m_0 = 0.05$ and $v = 64$ over interval $[0.053, 0.14]$. For the second topological layer, we use a superlevel set filtration on a V-constructed cubical complex with $v = 64$ over interval $[0.05, 0.75]$. We assign $\beta = 2\Delta t/\sqrt{\pi}$ to control the gradient intensity for ECLayr, and set $\lambda = 10000$ and $\lambda = 200$ for the first and second DECC layers, respectively. PersLay uses line point transform and a $10 \times 10$ unit grid for learnable weights over interval $[0.053, 0.14]^2$. $top2$ function is used as the permutation invariant operation for PLLay and PersLay. For all topological models, the post-processing layer $g_\theta$ is implemented as a single fully connected layer with output dimension 32.

The detailed experiment results are provided in Table 5. Additionally, we conduct an ablation study on the MNIST dataset without contamination, varying a single hyperparameter while keeping all others fixed. The test accuracy of the CNN + ECLayr model across various hyperparameter settings is presented in Table 6.

## B.4 ORBIT5K

**Training setup.** ORBIT5K is a synthetic point cloud dataset generated by simulating a dynamical system, and is often used as a TDA benchmark. This dataset contains 5000 samples with an equal proportion across the five labels. We randomly sample 1000 training data with equal class proportion, and use the remaining 4000 samples as the test set. 30% of the training data is further used as a validation set. The remaining training setup is identical to that of the MNIST experiment.

Table 6: Test accuracy of CNN + ECLayr on 1000 MNIST data for different choice of hyperparameters.

| Interval 1 $[T_{min}, T_{max}]$ | | | Interval 2 $[T_{min}, T_{max}]$ | | |
|---|---|---|---|---|---|
| $[0.053, 0.1]$ | $[0.053, 0.3]$ | $[0.053, 0.5]$ | $[0.05, 0.55]$ | $[0.05, 0.95]$ | $[0.25, 0.75]$ |
| 89.66 | 90.26 | 90.14 | 89.81 | 90.14 | 90.64 |
| $(\pm1.04)$ | $(\pm0.82)$ | $(\pm0.78)$ | $(\pm0.91)$ | $(\pm1.06)$ | $(\pm0.55)$ |
| Discretization $v$ | | | Gradient Control $\beta$ | | |
| 16 | 32 | 128 | 0.01 | 0.05 | 0.1 |
| 89.47 | 89.92 | 89.93 | 90.01 | 89.49 | 89.88 |
| $(\pm0.96)$ | $(\pm0.70)$ | $(\pm1.59)$ | $(\pm0.66)$ | $(\pm0.71)$ | $(\pm0.54)$ |

Table 7: Test accuracy of models trained on ORBIT5K data with different corruption and noise probability. For each corruption and noise probability, the best accuracy is highlighted in bold.

| Models | Corruption & Noise Probability | | | | |
|---|---|---|---|---|---|
| | **0.00** | **0.05** | **0.10** | **0.15** | **0.20** |
| CNN | **86.33** | 83.27 | 80.94 | 76.75 | 74.74 |
| | $(\pm4.32)$ | $(\pm3.86)$ | $(\pm3.82)$ | $(\pm4.30)$ | $(\pm4.63)$ |
| CNN + ECLayr(i) | **86.33** | 83.57 | 81.82 | 77.58 | 74.26 |
| | $(\pm2.43)$ | $(\pm3.85)$ | $(\pm3.29)$ | $(\pm5.39)$ | $(\pm3.33)$ |
| CNN + ECLayr | **86.33** | 84.13 | **82.41** | **80.43** | **77.81** |
| | $(\pm2.02)$ | $(\pm2.21)$ | $(\pm3.18)$ | $(\pm2.53)$ | $(\pm2.54)$ |
| CNN + DECC | 85.18 | **85.01** | 81.10 | 78.59 | 76.60 |
| | $(\pm3.03)$ | $(\pm2.95)$ | $(\pm3.28)$ | $(\pm3.82)$ | $(\pm2.90)$ |
| CNN + PersLay | 84.25 | 80.80 | 78.64 | 72.23 | 69.60 |
| | $(\pm2.04)$ | $(\pm3.51)$ | $(\pm4.17)$ | $(\pm3.01)$ | $(\pm3.90)$ |
| CNN + PLLay(i) | 84.06 | 80.33 | 77.17 | 70.70 | 69.59 |
| | $(\pm2.44)$ | $(\pm3.30)$ | $(\pm4.56)$ | $(\pm4.17)$ | $(\pm4.69)$ |
| CNN + PLLay | 83.90 | 79.66 | 76.94 | 71.88 | 72.70 |
| | $(\pm2.39)$ | $(\pm5.98)$ | $(\pm4.27)$ | $(\pm4.95)$ | $(\pm3.60)$ |

**Hyperparameter setting.** Both the first and second topological layers use a DTM filtration in equation 8 with $m_0 = 0.02$, where we define a $28 \times 28$ fixed grid on $[0.0125, 0.9875]^2$. $v = 16$ for both topological layers, and the filtration interval is set to $[0.075, 0.15]$ for the first layer and $[0.05, 0.15]$ for the second layer. We assign $\beta = 2\Delta t/\sqrt{\pi}$ to control the gradient intensity for ECLayr, and set $\lambda = 10000$ and $\lambda = 200$ for the first and second DECC layers, respectively. PersLay uses line point transform and a $10 \times 10$ unit grid for learnable weights over interval $[0.075, 0.15]^2$. $top2$ function is used as the permutation invariant operation for PLLay and PersLay. For all topological models, the post-processing layer $g_\theta$ is implemented as a single fully connected layer with output dimension 16.

The detailed experiment results are provided in Table 7. Additionally, we conduct an ablation study on the ORBIT5K dataset without contamination, varying a single hyperparameter while keeping all others fixed. The test accuracy of the CNN + ECLayr model across various hyperparameter settings is presented in Table 8.

Table 8: Test accuracy of CNN + ECLayr on 1000 ORBIT5K data for different choice of hyperparameters.

| Interval 1 $[T_{min}, T_{max}]$ | | | Interval 2 $[T_{min}, T_{max}]$ | | |
|:---:|:---:|:---:|:---:|:---:|:---:|
| $[0.075, 0.1]$ | $[0.075, 0.3]$ | $[0.075, 0.5]$ | $[0.05, 0.1]$ | $[0.05, 0.3]$ | $[0.05, 0.5]$ |
| 86.03 ($\pm$2.49) | 86.46 ($\pm$1.94) | 86.74 ($\pm$2.32) | 84.78 ($\pm$4.19) | 85.96 ($\pm$1.94) | 87.08 ($\pm$1.39) |
| Discretization $v$ | | | Gradient Control $\beta$ | | |
| 32 | 64 | 128 | 0.01 | 0.05 | 0.1 |
| 85.13 ($\pm$4.65) | 83.20 ($\pm$4.19) | 86.19 ($\pm$1.81) | 86.19 ($\pm$1.99) | 85.84 ($\pm$2.73) | 86.86 ($\pm$1.89) |

### B.5 MedMNIST3D

**Training setup.** MedMNIST3D (Yang et al., 2023) is a collection of 3D biomedical image datasets that encompasses diverse data modalities, including CTs, X-rays, MRAs, and more. To highlight the scalability of our proposed layer in high-dimensional settings, we conduct a series of experiments on several 3D voxel datasets from this collection. Following Yang et al. (2023), ResNet18 with 3D convolutions is used as our vanilla baseline model. We place ECLayr in parallel at the input (referred to as ResNet + ECLayr(i)), and add an additional topological layer after the first residual layer (referred to as ResNet + ECLayr, DECC). To reduce the computational cost of the ResNet + DECC model, ECLayr is used in place of DECC for the first topological layer, as this layer does not require backpropagation. A visualization of the model architecture is presented in Figure 7-(b). We use a batch size of 32 and train for at most 100 epochs using the Adam optimizer with learning rate 0.001. Early stopping is applied after 5 stagnant epochs without validation loss improvement, and cross-entropy loss is used for classification.

**Hyperparameter setting.** In all settings, the first topological layer uses a superlevel set filtration on a V-constructed cubical complex with $v = 32$ and interval $[0.05, 0.95]$. Similarly, the second topological layer always uses a superlevel set filtration on a V-constructed cubical complex with $v = 32$, but the filtration interval varies by dataset. We set the second interval as $[0.05, 0.85]$ for Fracture3D, Organ3D, and Vessel3D, and as $[0.15, 0.95]$ for Synapse3D. Across all settings, two fully connected layers with hidden dimension 64 and output dimension 32 is commonly used as the post-processing layer $g_\theta$. $\beta$ is always given as $2\Delta t/\sqrt{\pi}$ for ResNet + ECLayr, and $\lambda$ is always given as 200 for ResNet + DECC.

### B.6 Connection to Downstream Tasks

In this section, we evaluate how effectively the ECC captured by ECLayr supports downstream tasks. To this end, we consider the MNIST dataset under different noise levels and analyze (i) the average ECC for each class, (ii) the pairwise distance matrix among class-wise average ECCs, and (iii) top-3 closest classes for each class. Visualization of ECCs in Figure 8 show that in the noiseless case, the ECCs with similar topological structures have closer ECC values. For instance, digits with one loop (e.g., 0, 6, 9) have ECC values that are closer to zero compared to other digits. On the other hand, digits with no loops (e.g., 1, 7) have ECC values that are closer to one compared to other digits. Such trend is maintained up to approximately $10 \sim 15\%$ of noise, similar to our previous classification result. This observation is also supported by the heatmap in Figure 8, where digits with similar topological structures generally exhibit smaller pairwise distances between average ECCs. To facilitate the observation, Table 9 provides a summary of top-3 closest digits for each class and noise level. In general, digits with loops are closer in ECC space to other digits with loop structures, while digits without loops are closer to other digits without loops. Overall, the observations justify the utility of ECCs in downstream classification tasks.

Table 9: Top-3 classes closest in terms of distance between average ECCs.

| Digits | Corruption & Noise Probability | | | | |
| --- | --- | --- | --- | --- | --- |
| | 0.00 | 0.05 | 0.10 | 0.15 | 0.20 |
| 0 | 8,9,6 | 8,9,6 | 8,2,3 | 8,2,3 | 8,2,3 |
| 1 | 7,4,3 | 7,4,2 | 7,4,5 | 7,4,6 | 7,4,9 |
| 2 | 4,3,7 | 4,3,7 | 5,3,4 | 3,5,6 | 3,5,6 |
| 3 | 5,4,7 | 5,4,2 | 5,2,4 | 5,2,6 | 2,5,6 |
| 4 | 7,1,3 | 7,2,3 | 7,5,2 | 7,6,9 | 7,5,9 |
| 5 | 3,4,7 | 3,4,2 | 3,2,4 | 3,2,6 | 3,2,4 |
| 6 | 9,2,4 | 9,2,4 | 9,2,4 | 9,2,4 | 9,2,4 |
| 7 | 1,4,3 | 4,2,3 | 4,5,2 | 4,6,9 | 4,9,6 |
| 8 | 9,6,0 | 9,6,0 | 9,6,0 | 6,0,2 | 2,6,0 |
| 9 | 6,2,8 | 6,8,2 | 6,4,2 | 6,4,2 | 6,4,7 |

## C  More background on Topological Data Analysis

### C.1  Additional Definitions

We briefly review basic concepts in Topological Data Analysis that are needed to develop stability results in Section 5 of this paper, mainly coming from Kim et al. (2020). We refer interested readers to Chazal & Michel (2021); Hatcher (2002); Edelsbrunner & Harer (2010); Chazal et al. (2009; 2016a) for details and formal definitions.

**Vietoris-Rips Complex.** Let $X$ be a finite set of points in $\mathbb{R}^d$. For $r > 0$, the *Vietoris-Rips complex* is a collection of simplices where the distance between any two vertices is smaller than $2r$:

$$\text{Rips}(r) = \{\sigma \subset X \mid d(u_i, u_j) < 2r, \forall u_i, u_j \in \sigma\}.$$

Notice that $\text{Rips}(r_1) \subset \text{Rips}(r_2)$ when $r_1 \leq r_2$. Thus, we can build a filtration on the Vietoris-Rips complex by monotonically increasing $r$.

**Alpha Complex.** Let $X$ be a finite set of points in $\mathbb{R}^d$. For each $u_i \in X$, the *Voronoi cell* of $u_i$ is the set of points that are closest to $u_i$; $V_{u_i} = \{x \in \mathbb{R}^d | d(u_i, x) \leq d(u_j, x), \forall u_j \in X, u_j \neq u_i\}$. For $r > 0$ and each $u_i \in X$, let us denote the closed *r-ball* with center $u_i$ and radius $r$ as $B_{u_i}(r)$. Then, we define $R_{u_i}(r) = B_{u_i}(r) \cap V_{u_i}$, which is the intersection of each $r$-ball with its corresponding Voronoi cell. The *Alpha complex* is a collection of simplices such that all $R_{u_i}(r)$ of the vertices in the simplex have an intersection:

$$\text{Alpha}(r) = \{\sigma \subset X | \cap_{u_i \in \sigma} R_{u_i}(r) \neq \emptyset\}.$$

Similar to the Vietoris-Rips complex, we can build a filtration on the Alpha complex by monotonically increasing $r$.

**Wasserstein Distance.** We suggest two versions of Wasserstein distances, one is for persistence diagrams and the other is for probability measures.

We first start with Wasserstein distance for persistence diagrams. A *matching* between two persistence diagrams $\mathcal{D}_1$ and $\mathcal{D}_2$, is a subset $m \subset \mathcal{D}_1 \times \mathcal{D}_2$ such that every off-diagonal point in $\mathcal{D}_1$ and $\mathcal{D}_2$ only appears once in $m$. The *p-Wasserstein distance* between persistence diagrams is defined by

$$W_p(\mathcal{D}_1, \mathcal{D}_2) = \inf_{\text{matching } m} \left( \sum_{(x,y) \in m} \|x - y\|_\infty^p \right)^{1/p}$$

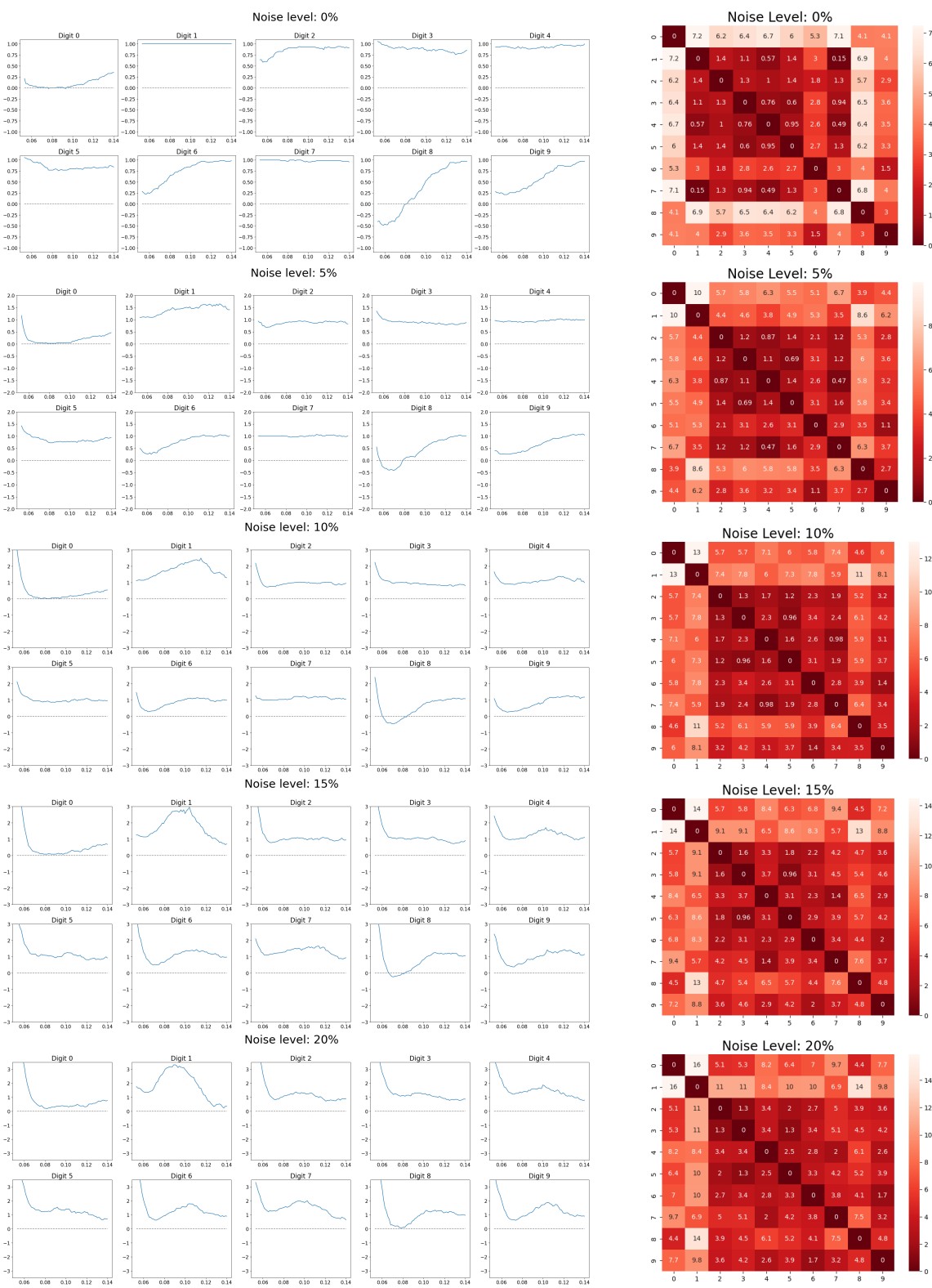

Figure 8: (Left) Average ECCs of each class in MNIST per noise level. (Right) Heatmap of the pairwise distance matrix of average ECCs per noise level.

Now we see Wasserstein distance for probability measures. Let $P$ and $Q$ be probability measures on $\mathcal{X}$, and let $\mathcal{J}(P, Q)$ denote all joint distributions $J$ for $\mathcal{X} \times \mathcal{X}$ that have marginals $P$ and $Q$. In other words, $(\Pi_1)_\# J = P$ and $(\Pi_2)_\# J = Q$ where $\Pi_1(x, y) = x$ and $\Pi_2(x, y) = y$, and $T_\# P$ is a push-forward measure of $P$, i.e., $T_\# P(A) = P\left(\{x : T(x) \in A\right) = P(T^{-1}(A))$. For $p \geq 1$, the Kantorovich, or Wasserstein, distance is

$$W_p(P, Q) = \left(\inf_{J \in \mathcal{J}(P,Q)} \int_{\mathcal{X} \times \mathcal{X}} ||x - y||^p dJ(x, y)\right)^{1/p}.$$

**Gromov-Hausdorff distance.** The *Hausdorff distance* is on sets embedded in the same metric spaces. This distance measures how two sets are close to each other in the embedded metric space. When $S \subset \mathbb{X}$, we denote by $S^r$ the $r$-neighborhood of a set $S$ in $\mathbb{R}^d$, i.e. $S^r = \bigcup_{x \in S} B_x(r)$.

**Definition C.1** (Hausdorff distance (Burago et al., 2001, Definition 7.3.1)). *Let $X, Y \subset \mathbb{X}$ be subsets of $\mathbb{R}^d$. The* Hausdorff distance *between $X$ and $Y$, denoted by $d_H(X, Y)$, is defined as*

$$d_H(X, Y) := \inf\{r > 0 : X \subset Y^r \text{ and } Y \subset X^r\}.$$

The notion of the Hausdorff distance can be generalized to the comparison of any pair of metric spaces. The *Gromov-Hausdorff distance* measures how two sets are far from being isometric to each other.

**Definition C.2** (Gromov-Hausdorff distance (Burago et al., 2001, Definition 7.3.10)). *Let $X$ and $Y$ be two metric spaces. The* Gromov-Hausdorff distance *between $X$ and $Y$, denoted by $d_{GH}(X, Y)$, is defined as*

$$d_{GH}(X, Y) := \inf\{d_H(X', Y') : \text{ there exists a metric space}$$
$$Z \text{ and } X', Y' \subset Z \text{ with } X, Y \text{ isometric to}$$
$$X', Y', \text{ respectively.}\}$$

**Distance to measure.** Distance to measure (DTM) (Chazal et al., 2011; 2016b; Anai et al., 2020) is a distance-like function[2] that is robust to outliers. For a probability measure $\mu$ and parameters $m_0 \in (0, 1)$ and $r \geq 1$, the DTM function $d_{\mu,m_0} : \mathbb{R}^d \to \mathbb{R}$ is defined as

$$d_{\mu,m_0}(x) = \left(\frac{1}{m_0} \int_0^{m_0} (\delta_{\mu,m}(x))^r dm\right)^{1/r},$$

where $\delta_{\mu,m}(x) = \inf\{t > 0 \mid \mu(B_x(t)) > m\}$ and $B_x(t)$ is a closed $t$-ball centered at $x$.

In practice, an empirical DTM is used (see Figure 3-(b)). If input data X is considered as weights corresponding to fixed points Y,

$$\hat{d}_{m_0}(x) = \left(\frac{\sum_{Y_i \in N_k(x)} X_i' \|Y_i - x\|^r}{m_0 \sum_{i=1}^n X_i}\right)^{1/r}, \tag{7}$$

where $N_k(x)$ is a subset of $Y$ containing the $k$ nearest neighbors of $x$. $k$ is such that satisfies $\sum_{Y_i \in N_{k-1}(x)} X_i < m_0 \sum_{i=1}^n X_i \leq \sum_{Y_i \in N_k(x)} X_i$, and $X_i' = \sum_{Y_j \in N_k(x)} X_j - m_0 \sum_{j=1}^n X_j$ if at least one of $Y_i$'s is in $N_k(x)$ and $X_i' = X_i$ otherwise.

When input data is considered as empirical data points, the empirical DTM becomes

$$\hat{d}_{m_0}(x) = \left(\frac{\sum_{X_i \in N_k(x)} w_i' \|X_i - x\|^r}{m_0 \sum_{i=1}^n w_i}\right)^{1/r}, \tag{8}$$

where $N_k(x)$ is a subset of $X$ containing the $k$ nearest neighbors of $x$. $k$ is such that satisfies $\sum_{X_i \in N_{k-1}(x)} w_i < m_0 \sum_{i=1}^n w_i \leq \sum_{X_i \in N_k(x)} w_i$, and $w_i' = \sum_{X_j \in N_k(x)} w_j - m_0 \sum_{j=1}^n w_j$ if at least one of $X_i$'s is in $N_k(x)$ and $w_i' = w_i$ otherwise.

---

[2]This distance function is not the distance function giving a metric between two input points such as $L_p$ distance, but rather measures a distance between a single input point and the support set of a probability distribution.

The parameter $m_0$ determines how much local or global structures should be extracted, with smaller $m_0$ corresponding to more local structures. The DTM function is differentiable (Kim et al., 2020), and adopting a sublevel or superlevel set filtration on the DTM transformed data yields a DTM filtration that is robust to outliers.

### C.2 Constructing Filtered Cubical Complexes from Image Data

Let $X \in \mathbb{R}^{H \times W}$ be a 2D image. There are two methods of constructing a filtered cubical complex: T-construction and V-construction.

**T-construction** In *T-construction*, each pixel in the image is mapped to a top-dimensional cell in the cubical complex, which is a square in case of 2D images. The filtration value of each square is assigned as the intensity of its corresponding pixel, and these filtration values are recursively extended to lower dimensional cubes. The filtration value of each edge is assigned as the minimum of the filtration values of its neighboring squares. Similarly, the filtration value of each vertex is assigned as the minimum of the filtration values its neighboring edges.

**V-construction** In *V-construction*, each pixel in the image is mapped to a vertex in the cubical complex. The filtration value of each vertex is assigned as the intensity of its corresponding pixel, and these filtration values are recursively extended to higher dimensional cubes. The filtration value of each edge is assigned as the maximum of the filtration values of its neighboring vertices. Similarly, the filtration value of each square is assigned as the maximum of the filtration values of its neighboring edges.

For both constructions, a sublevel set at a given filtration value $t$ defines a subcomplex $K(t) := \{\sigma \in K | f(\sigma) \le t\}$; the collection of cubes with filtration value less than or equal to $t$. Consequently, a sublevel set filtration can be built by monotonically increasing $t$. A superlevel set filtration can also be obtained by applying the sublevel set filtration to a cubical complex constructed from $-X$ rather than $X$.

# D  Derivative of Filtration value with respect to Input $X$: $\frac{\partial f(\sigma)}{\partial X}$

### D.1  Vietoris-Rips Filtration

Assume Vietoris-Rips general position for a point cloud $X$: (i) all points in $X$ are unique, and (ii) the length of all attaching edges are unique. The filtration value of a simplex $\sigma$ in the Vietoris-Rips filtration is half the length of the longest edge in $\sigma$. This edge is the *attaching edge* of $\sigma$, denoted as $\tau_\sigma$. Letting $x_i$ and $x_j$ be the vertices of $\tau_\sigma$, the derivatives of filtration value $f(\sigma) = \frac{\|x_i - x_j\|}{2}$ with respect to the points $x_i$ and $x_j$ are given by [25]:

$$\frac{\partial f(\sigma)}{\partial x_i} = \frac{1}{2} \frac{x_i - x_j}{\|x_i - x_j\|}, \quad \frac{\partial f(\sigma)}{\partial x_j} = \frac{1}{2} \frac{x_j - x_i}{\|x_i - x_j\|}. \tag{9}$$

The derivatives with respect to points other than $x_i$ and $x_j$ are all zero.

### D.2  Alpha Filtration

Assume Alpha general position of a point cloud $X$: (i) general position in the sense of [23], and (ii) filtration values of all attaching simplices are unique. In Alpha filtration, all simplices are either an attaching simplex, or a simplex attached by another simplex of higher dimension. In the latter case, filtration value of the attached simplex is given by the filtration value of its attaching simplex. The filtration value of an attaching simplex $\sigma$ is the radius of the smallest circumcircle of $\sigma$ [23], [25] and it can be differentiated with respect to the coordinates of each of the vertices.

### D.3  Sub/Superlevel Set Filtration on Filtered Cubical Complexes

Let us treat a 2D image $X \in \mathbb{R}^{H \times W}$ as a vector $x = (x_1, \ldots, x_{HW}) \in \mathbb{R}^{HW}$, where the elements of the vector are arranged in row-major order. Then, the derivative of the filtration value with respect to the input data

can be written as

$$\frac{\partial f(\sigma)}{\partial x} = \left( \frac{\partial f(\sigma)}{\partial x_1}, \ldots, \frac{\partial f(\sigma)}{\partial x_{HW}} \right)$$

Given that the filtration value varies depending on the construction used, we provide differentiability results for both T-construction and V-constructions. For simplicity of notation, we denote $\mathcal{I} = \{1, 2, \ldots, HW\}$ as the index set.

**T-construction.** In T-construction, each pixel is mapped to a square, with the pixel intensity serving as the filtration value of the corresponding square. Thus, we first explore the scenario where $\sigma$ is a square, and then extend our analysis to lower dimensional cubes.

(i) Assume $\sigma$ is a square, i.e., $\dim(\sigma) = 2$. Let $j \in \mathcal{I}$ denote the index of the pixel in $x$ that corresponds to $\sigma$. Then, $f(\sigma) = x_j$ and thus,

$$\frac{\partial f(\sigma)}{\partial x_i} = \begin{cases} 1, & \text{if } i = j \\ 0, & \text{otherwise} \end{cases}$$

for all $i \in \mathcal{I}$.

(ii) Assume $\sigma$ is an edge, i.e., $\dim(\sigma) = 1$. Recall that $f(\sigma)$ is assigned as the minimum filtration value of its neighboring squares, which in turn is equivalent to the minimum pixel intensity of the pixels corresponding to those neighboring squares. Thus, we can identify the pixel associated with $\sigma$ by

1. find neighboring squares of $\sigma$

2. determine the neighboring square with minimum filtration value

3. identify the pixel that corresponds to the square found in (2)

In step 2, multiple neighboring squares may have the same minimum filtration value. In this case, we identify the set of pixels that corresponds to all such squares. Letting $J \subset \mathcal{I}$ denote an index set labeling the members of such set of pixels,

$$\frac{\partial f(\sigma)}{\partial x_i} = \begin{cases} 1/|J|, & \text{if } i \in J \\ 0, & \text{otherwise} \end{cases}$$

for all $i \in \mathcal{I}$. Observe that when multiple pixels contribute to $\sigma$, we distribute the gradient evenly between those pixels.

(iii) Assume $\sigma$ is a vertex, i.e., $\dim(\sigma) = 0$. Recall that $f(\sigma)$ is assigned as the minimum filtration value of its neighboring edges. Therefore, once we find the neighboring edge(s) with minimum filtration value, we can repeat the process in (ii) to identify the set of pixels associated with $\sigma$. Letting $J \subset \mathcal{I}$ denote an index set labeling the members of such set of pixels,

$$\frac{\partial f(\sigma)}{\partial x_i} = \begin{cases} 1/|J|, & \text{if } i \in J \\ 0, & \text{otherwise} \end{cases}$$

for all $i \in \mathcal{I}$.

**V-construction.** In V-construction, each pixel is mapped to a vertex, with the pixel intensity serving as the filtration value of the corresponding vertex. Thus, we first explore the scenario where $\sigma$ is a vertex, and then extend our analysis to higher dimensional cubes.

(i) Assume $\sigma$ is a vertex, i.e., $\dim(\sigma) = 0$. Let $j \in \mathcal{I}$ be the index of the pixel in $x$ that corresponds to $\sigma$. Then, $f(\sigma) = x_j$ and thus,

$$\frac{\partial f(\sigma)}{\partial x_i} = \begin{cases} 1, & \text{if } i = j \\ 0, & \text{otherwise} \end{cases}$$

for all $i \in \mathcal{I}$.

(ii) Assume $\sigma$ is an edge, i.e., $\dim(\sigma) = 1$. Recall that $f(\sigma)$ is assigned as the maximum filtration value of its neighboring vertices, which in turn is equivalent to the maximum pixel intensity of the pixels corresponding to those neighboring vertices. Thus, we can identify the pixel associated with $\sigma$ by

1. find neighboring vertices of $\sigma$

2. determine the neighboring vertex with maximum filtration value

3. identify the pixel that corresponds to the vertex found in (2)

In step 2, multiple neighboring vertices may have the same maximum filtration value. In this case, we identify the set of pixels that corresponds to all such vertices. Letting $J \subset \mathcal{I}$ denote an index set labeling the members of such set of pixels,

$$\frac{\partial f(\sigma)}{\partial x_i} = \begin{cases} 1/|J|, & \text{if } i \in J \\ 0, & \text{otherwise} \end{cases}$$

for all $i \in \mathcal{I}$. Observe that when multiple pixels contribute to $\sigma$, we distribute the gradient evenly between those pixels.

(iii) Assume $\sigma$ is a square, i.e., $\dim(\sigma) = 2$. Recall that $f(\sigma)$ is assigned as the maximum filtration value of its neighboring edges. Therefore, once we find the neighboring edge(s) with maximum filtration value, we can repeat the process in (ii) to identify the set of pixels associated with $\sigma$. Letting $J \subset \mathcal{I}$ denote an index set labeling the members of such set of pixels,

$$\frac{\partial f(\sigma)}{\partial x_i} = \begin{cases} 1/|J|, & \text{if } i \in J \\ 0, & \text{otherwise} \end{cases}$$

for all $i \in \mathcal{I}$.

## E   Additional Stability Analysis

When $K = K'$, $\|f_X - f_{X'}\|_\infty = \sup_{\sigma \in K} |f_X(\sigma) - f_{X'}(\sigma)|$, and if $K \neq K'$, $\|f_X - f_{X'}\|_\infty$ is not well defined. However, there is a general distance between two filtration functions $f_X$ and $f_{X'}$ even when base simplicial complexes (or cubical complexes) are different; it is the interleaving distance $d_I(f_X, f_{X'})$. For the definition, see Section 5.1 from Chazal et al. (2016a). When $K = K'$, there is a bound

$$d_I(f_X, f_{X'}) \leq \|f_X - f_{X'}\|_\infty. \tag{10}$$

Hence, we have a general version of Theorem 5.2 as follows.

**Theorem E.1.** *Suppose $K$ is a finite simplicial complex or cubical complex. Then there exists a constant $C_K$ only depending on $K$ such that*

$$\|\mathcal{C}_X - \mathcal{C}_{X'}\|_1 \leq C_K d_I(f_X, f_{X'}).$$

**Corollary E.2.** *Suppose $f_X$, $f_{X'}$ are Vietoris-Rips filtrations of $X$ and $X'$, respectively. Then*

$$\|\mathcal{C}_X - \mathcal{C}_{X'}\|_1 \leq C_K d_{GH}(X, X'),$$

*where $d_{GH}$ is the Gromov-Hausdorff distance.*

## F   Proofs

### F.1   Proofs of Section 3

*Proof of Proposition 3.1.* We prove the two parts in turn. Throughout, we interpret $T(K(T_{\max}), v)$ as the runtime of Algorithm 1 when the input complex is the *processed* subcomplex $K(T_{\max}) = \{\sigma \in K : f(\sigma) \leq T_{\max}\}$, as stated in the proposition.

*(i) Deterministic bound.* On input $K(T_{\max})$, Algorithm 1 performs: (a) a single pass over all cells $\sigma \in K(T_{\max})$, (b) for each such $\sigma$, it computes the bin index $t^*$ and applies a constant-time update $\mathcal{E}(t^*) \leftarrow \mathcal{E}(t^*) + (-1)^{\dim(\sigma)}$, and (c) one cumulative sum over the $v$ grid entries.

Because $t_{\text{seq}} = \{t_1, \ldots, t_v\}$ is evenly spaced on $[T_{\min}, T_{\max}]$, the index $t^*$ can be computed in $O(1)$ time from $f(\sigma)$ by arithmetic. Concretely, letting $h = (T_{\max} - T_{\min})/(v-1)$ and defining

$$i(\sigma) = \min\Big\{v, \ \max\big\{1, \ \lceil (f(\sigma) - T_{\min})/h \rceil + 1\big\}\Big\},$$

we have $t^* = t_{i(\sigma)}$, and evaluating $i(\sigma)$ also requires constant work (i.e., $O(1)$ per $\sigma$). Therefore, there exist constants $a, b > 0$ such that

$$T(K(T_{\max}), v) \ \leq \ a\,|K(T_{\max})| + b\,v.$$

*(ii) Expectation.* By definition,

$$|K(T_{\max})| = \sum_{\sigma \in K} Z_\sigma, \qquad Z_\sigma = \mathbb{1}\{f(\sigma) \leq T_{\max}\}.$$

Taking expectations and using linearity,

$$\mathbb{E}|K(T_{\max})| = \sum_{\sigma \in K} \mathbb{E}Z_\sigma = \sum_{k \geq 0} \sum_{\sigma \in K^k} \mathbb{P}(Z_\sigma = 1) = \sum_{k \geq 0} p_k\,N_k$$

Now take expectations in the deterministic bound from part (i) to get

$$\mathbb{E}\,T(K(T_{\max}), v) \ \leq \ a\,\mathbb{E}|K(T_{\max})| + b\,v \ = \ a\sum_k p_k N_k + b\,v.$$

$\square$

## F.2 Proofs of Section 4

*Proof of Proposition 4.1.* First, note that the sigmoid function $S(x) = \frac{1}{1+\exp(-x)}$ satisfies

$$\frac{d}{dx}S(x) = \frac{\exp(-x)}{(1+\exp(-x))^2} = S(x)(1 - S(x)),$$

and hence

$$\frac{\partial S(\lambda(t - f(\sigma)))}{\partial f(\sigma)} = -\lambda S(\lambda(t - f(\sigma)))(1 - S(\lambda(t - f(\sigma)))).$$

Meanwhile,

$$\left|\frac{\partial S(\lambda(t - f(\sigma)))}{\partial f(\sigma)}\bigg|_{t=t_i}\right| = \lambda S(\lambda(t_i - f(\sigma)))\left[1 - S(\lambda(t_i - f(\sigma)))\right]$$
$$\leq \lambda S(\lambda d(f(\sigma), tseq))\left[1 - S(\lambda d(f(\sigma), tseq))\right].$$

Hence,

$$\left\|S'^{tseq}_{\lambda, f(\sigma)}\right\|_\infty = \left\|\frac{\partial S(\lambda(t - f(\sigma)))}{\partial f(\sigma)}\big|_{t=t_1, \ldots, t_v}\right\|_\infty$$
$$\leq \lambda S(\lambda d(f(\sigma), tseq))\left[1 - S(\lambda d(f(\sigma), tseq))\right],$$

where $d(f(\sigma), tseq)$ denotes the distance from $f(\sigma)$ to its nearest grid point. Thus, if $f(\sigma) = t_i - \Delta t$ for some $t_i \in tseq$, then

$$\left\|S'^{tseq}_{\lambda, f(\sigma)}\right\|_\infty \leq \lambda S(\lambda \Delta t)\left[1 - S(\lambda \Delta t)\right].$$
$$\leq \lambda \exp(-\lambda \Delta t).$$

If either (i) $\Delta t$ is fixed and $\lambda \to \infty$, or (ii) $\lambda$ is fixed and $\Delta t \to \infty$, such that $\lambda \exp(-\lambda \Delta t) \to 0$, then

$$\inf_{f(\sigma) \in [t_1 - \Delta t, t_v + \Delta t]} \left\| S'^{tseq}_{\lambda, f(\sigma)} \right\|_\infty \to 0.$$

$\square$

*Proof of Proposition 4.2.* $\delta^{tseq}_{\beta, f(\sigma)}$ always has the form

$$\left( 0, \ldots, 0, -\frac{\alpha}{\beta\sqrt{\pi}}, -\frac{1-\alpha}{\beta\sqrt{\pi}}, 0, \ldots, 0 \right),$$

where $0 \le \alpha = \frac{t_i - f(\sigma)}{2\Delta t} \le 1$. Therefore,

$$\left\| \hat{\delta}^{tseq}_{\beta, f(\sigma)} \right\|_\infty = \max \left( \frac{\alpha}{\beta\sqrt{\pi}}, \frac{1-\alpha}{\beta\sqrt{\pi}} \right)$$

has minimum value $\frac{1}{2\beta\sqrt{\pi}}$ when $\alpha = 1/2$. $\square$

Before beginning the proofs of Propositions 4.3 and 4.4, we would first like to emphasize that for all $t \in [t_1 - \Delta t, t_v + \Delta t)$,

$$\int_{t_1 - \Delta t}^{t_v + \Delta t} \frac{\partial \mathbb{1}(f(\sigma) \le t)}{\partial f(\sigma)} df(\sigma) = \mathbb{1}(t_v + \Delta t \le t) - \mathbb{1}(t_1 - \Delta t \le t)$$

$$= -1,$$

and for all $f(\sigma) \in [t_1 - \Delta t, t_v + \Delta t)$,

$$\int_{t_1 - \Delta t}^{t_v + \Delta t} \frac{\partial \mathbb{1}(f(\sigma) \le t)}{\partial f(\sigma)} dt = -\int_{t_1 - \Delta t}^{t_v + \Delta t} \frac{\partial \mathbb{1}(f(\sigma) \le t)}{\partial t} dt$$

$$= -\mathbb{1}(f(\sigma) \le t_v + \Delta t) + \mathbb{1}(f(\sigma) \le t_1 - \Delta t)$$

$$= -1.$$

*Proof of Proposition 4.3.* For given $t \in [t_1 - \Delta t, t_v + \Delta t)$, let $t_i \in tseq$ be such that $t \in [t_i - \Delta t, t_i + \Delta t)$. Then

$$\int_{t_1 - \Delta t}^{t_v + \Delta t} \hat{S}'_{\lambda, f(\sigma)}(t) df(\sigma) = \int_{t_1 - \Delta t}^{t_v + \Delta t} \frac{\partial S(\lambda(t_i - f(\sigma)))}{\partial f(\sigma)} df(\sigma)$$

$$= S(\lambda(t_i - t_v - \Delta t)) - S(\lambda(t_i - t_1 + \Delta t)).$$

This is minimized when $t_i$ is close to $\frac{t_1 + t_v}{2}$. Thus, we have

$$\int_{t_1 - \Delta t}^{t_v + \Delta t} \hat{S}'_{\lambda, f(\sigma)}(t) df(\sigma) \ge S(-\lambda v \Delta t) - S(\lambda v \Delta t),$$

and hence

$$\left| \int_{t_1 - \Delta t}^{t_v + \Delta t} \hat{S}'_{\lambda, f(\sigma)}(t) df(\sigma) - \int_{t_1 - \Delta t}^{t_v + \Delta t} \frac{\partial \mathbb{1}(f(\sigma) \le t)}{\partial f(\sigma)} df(\sigma) \right|$$

$$\ge 1 - S(\lambda v \Delta t) + S(-\lambda v \Delta t)$$

$$= 2S(-\lambda v \Delta t).$$

Also, note that from the calculation in the proof of Proposition 4.1, if $f(\sigma) = t_i - \Delta t$ for some $t_i \in tseq$, then

$$\left\| S'^{tseq}_{\lambda, f(\sigma)} \right\|_\infty \leq \lambda S(\lambda \Delta t) \left[ 1 - S(\lambda \Delta t) \right].$$
$$\leq \lambda \exp(-\lambda \Delta t).$$

And

$$\left| \int_{t_1 - \Delta t}^{t_v + \Delta t} \hat{S}'_{\lambda, f(\sigma)}(t) dt \right| \leq \left\| S'^{tseq}_{\lambda, f(\sigma)} \right\|_\infty 2v\Delta t \leq 2\lambda v \Delta t \exp(-\lambda \Delta t).$$

Therefore,

$$\left| \int \hat{S}'_{\lambda, f(\sigma)}(t) dt - \int \frac{\partial \mathbb{1}(f(\sigma) \leq t)}{\partial f(\sigma)} dt \right|$$
$$\geq \left| 1 - 2\lambda v \Delta t \exp(-\lambda \Delta t) \right|.$$

$\square$

*Proof of Proposition 4.4.* For given $t \in [t_1 - \Delta t, t_v + \Delta t)$, let $t_i \in tseq$ be such that $t \in [t_i - \Delta t, t_i + \Delta t)$. Then $\hat{\delta}_{\beta, f(\sigma)}(t)$ is nonzero if and only if $f(\sigma) \in [t_{i-1}, t_{i+1})$, and hence

$$\int_{t_1 - \Delta t}^{t_v + \Delta t} \hat{\delta}_{\beta, f(\sigma)}(t) df(\sigma) = \int_{t_{i-1}}^{t_i} -\left( 1 - \frac{t_i - f(\sigma)}{2\Delta t} \right) \frac{1}{\beta \sqrt{\pi}} df(\sigma) + \int_{t_i}^{t_{i+1}} -\left( \frac{t_{i+1} - f(\sigma)}{2\Delta t} \right) \frac{1}{\beta \sqrt{\pi}} df(\sigma)$$

$$= \frac{t_i - 2\Delta t}{\beta \sqrt{\pi}} - \frac{1}{2\Delta t \beta \sqrt{\pi}} \int_{t_{i-1}}^{t_i} f(\sigma) df(\sigma) - \frac{t_{i+1}}{\beta \sqrt{\pi}} + \frac{1}{2\Delta t \beta \sqrt{\pi}} \int_{t_i}^{t_{i+1}} f(\sigma) df(\sigma)$$

$$= -\frac{4\Delta t}{\beta \sqrt{\pi}} - \frac{t_i^2 - t_{i-1}^2}{4\Delta t \beta \sqrt{\pi}} + \frac{t_{i+1}^2 - t_i^2}{4\Delta t \beta \sqrt{\pi}}$$

$$= -\frac{4\Delta t}{\beta \sqrt{\pi}} - \frac{1}{4\Delta t \beta \sqrt{\pi}} \{ (t_i - t_{i-1})(t_i + t_{i-1}) - (t_{i+1} - t_i)(t_{i+1} + t_i) \}$$

$$= -\frac{4\Delta t}{\beta \sqrt{\pi}} - \frac{1}{4\Delta t \beta \sqrt{\pi}} \{ 2\Delta t(2t_i - 2\Delta t) - 2\Delta t(2t_i + 2\Delta t) \}$$

$$= -\frac{4\Delta t}{\beta \sqrt{\pi}} + \frac{2\Delta t}{\beta \sqrt{\pi}}$$

$$= -\frac{2\Delta t}{\beta \sqrt{\pi}},$$

since $t_i = t_{i-1} + 2\Delta t = t_{i+1} - 2\Delta t$.

And for given $f(\sigma) \in [t_1 - \Delta t, t_v + \Delta t)$, let $t_i \in tseq$ be such that $f(\sigma) \in [t_{i-1}, t_i)$. Then $\hat{\delta}_{\beta, f(\sigma)}(t)$ is nonzero if and only if $t \in [t_{i-1} - \Delta t, t_i + \Delta t)$, and hence

$$\int_{t_1 - \Delta t}^{t_v + \Delta t} \hat{\delta}_{\beta, f(\sigma)}(t) dt = \int_{t_{i-1} - \Delta t}^{t_{i-1} + \Delta t} -\left( \frac{t_i - f(\sigma)}{2\Delta t} \right) \frac{1}{\beta \sqrt{\pi}} dt + \int_{t_i - \Delta t}^{t_i + \Delta t} -\left( 1 - \frac{t_i - f(\sigma)}{2\Delta t} \right) \frac{1}{\beta \sqrt{\pi}} dt$$

$$= -\frac{t_i - f(\sigma)}{\beta \sqrt{\pi}} + \frac{t_i - f(\sigma) - 2\Delta t}{\beta \sqrt{\pi}}$$

$$= -\frac{2\Delta t}{\beta \sqrt{\pi}}.$$

$\square$

## F.3 Proofs of Section 5

*Proof of Proposition 5.1.* Since $\mathcal{O}_\theta(X) = g_\theta(\mathcal{C}_X(tseq))$ and $\mathcal{O}_\theta(X') = g_\theta(\mathcal{C}_{X'}(tseq))$,

$$\| \mathcal{O}_\theta(X) - \mathcal{O}_\theta(X') \|_1 = \| g_\theta(\mathcal{C}_X(tseq)) - g_\theta(\mathcal{C}_{X'}(tseq)) \|_1$$
$$\leq L \| \mathcal{C}_X(tseq) - \mathcal{C}_{X'}(tseq) \|_1.$$

Now, note that the ECC $\mathcal{C}_X$ can be expanded using persistence diagrams $\{\mathcal{D}_k(X) : k \geq 0\}$ as follows: if $\mathcal{D}_k(X) = \{(b_{ki}, d_{ki}) : 1 \leq i \leq n_k\}$, then by definition,

$$\mathcal{C}_X(t) = \sum_{k=0}^{\infty} (-1)^k \sum_{i=1}^{n_k} \mathbb{1}(b_{ki} \leq t < d_{ki}).$$

Since $b_{ki}, d_{ki} \in \{t_i^*\}_{i=1}^w$, the value of $\mathcal{C}_X(t) - \mathcal{C}_{X'}(t)$ only changes at critical points $t_i^*$, and $\mathcal{C}_X(t) - \mathcal{C}_{X'}(t)$ is therefore a piecewise constant function that takes a constant value on every disjoint interval $[t_i^*, t_{i+1}^*)$. Hence, there exists $a_1, \ldots, a_{w-1} \in \mathbb{Z}$ such that $\mathcal{C}_X(t) - \mathcal{C}_{X'}(t)$ can be expressed as

$$\mathcal{C}_X(t) - \mathcal{C}_{X'}(t) = \sum_{i=1}^{w-1} a_i \mathbb{1}(t_i^* \leq t < t_{i+1}^*).$$

Then from $tseq = \{t_1, \ldots, t_v\}$, $\|\mathcal{C}_X(tseq) - \mathcal{C}_{X'}(tseq)\|_1$ is expanded as

$$\|\mathcal{C}_X(tseq) - \mathcal{C}_{X'}(tseq)\|_1 = \sum_{j=1}^{v} \left| \sum_{i=1}^{w-1} a_i \mathbb{1}(t_i^* \leq t_j < t_{i+1}^*) \right|$$

$$= \sum_{j=1}^{v} \sum_{i=1}^{w-1} |a_i| \, \mathbb{1}(t_i^* \leq t_j < t_{i-1}^*),$$

where the second equality follows from the fact that the intervals $\{[t_i^*, t_{i+1}^*)\}_{i=1}^{w-1}$ are mutually disjoint. Similarly, $\|\mathcal{C}_X - \mathcal{C}_{X'}\|_1$ is expanded as

$$\|\mathcal{C}_X - \mathcal{C}_{X'}\|_1 = \int \left| \sum_{i=1}^{w-1} a_i \mathbb{1}(t_i^* \leq t < t_{i+1}^*) \right| dt$$

$$= \sum_{i=1}^{w-1} |a_i| \, (t_{i+1}^* - t_i^*).$$

Now for each $i = 1, \ldots, w-1$, $\sum_{j=1}^v \mathbb{1}(t_i^* \leq t_j < t_{i+1}^*)$ is the number of $t_j$'s that falls within the interval $[t_i^*, t_{i+1}^*)$. But since $t_{j+1} - t_j \geq \Delta t$, such number is at most $\left\lceil \frac{t_{i+1}^* - t_i^*}{\Delta t} \right\rceil$, and also from $2(t_{i+1}^* - t_i^*) \geq \Delta t$,

$$\sum_{j=1}^{v} \mathbb{1}(t_{n_{2i}}^* \leq t_j < t_{n_{2i+1}}^*) \leq \left\lceil \frac{t_{i+1}^* - t_i^*}{\Delta t} \right\rceil$$

$$\leq \frac{2(t_{i+1}^* - t_i^*)}{\Delta t}.$$

Hence $\|\mathcal{C}_X(tseq) - \mathcal{C}_{X'}(tseq)\|_1$ can be correspondingly upper bounded as

$$\|\mathcal{C}_X(tseq) - \mathcal{C}_{X'}(tseq)\|_1 = \sum_{i=1}^{w-1} |a_i| \sum_{j=1}^{v} \mathbb{1}(t_i^* \leq t_j \leq t_{i+1}^*)$$

$$\leq \frac{2}{\Delta t} \sum_{i=1}^{m} |a_i| \, (t_{i+1}^* - t_i^*)$$

$$= \frac{2}{\Delta t} \|\mathcal{C}_X - \mathcal{C}_{X'}\|_1.$$

And correspondingly,

$$\|\mathcal{O}_\theta(X) - \mathcal{O}_\theta(X')\|_1 \leq \frac{2L}{\Delta t} \|\mathcal{C}_X - \mathcal{C}_{X'}\|_1.$$

$\square$

*Proof of Theorem 5.2.* From Theorem E.1 , we have

$$\|\mathcal{C}_X - \mathcal{C}_{X'}\|_1 \leq C_K d_I(f_X, f_{X'}).$$

When $K = K'$, the additional bound in equation 10 yields

$$\|\mathcal{C}_X - \mathcal{C}_{X'}\|_1 \leq C_K \|f_X - f_{X'}\|_\infty.$$

□

*Proof of Corollary 5.3.* Before starting the proof, we first explain what an empirical distribution and a "restriction of a DTM function" mean. We say $P_X$ is an empirical distribution on $X = \{X_1, \ldots, X_n\}$, when $P_X = \frac{1}{n} \sum_{i=1}^n \delta_{X_i}$, where $\delta_{X_i}$ is a Dirac measure on $X_i$, i.e., $\delta_{X_i}(A) = \mathbb{1}(X_i \in A)$. And suppose $\{\sigma_i\} \subset K$ be vertices of $K$ for V-construction, or top dimensional cells of $K$ for T-construction. Then we say $f_X$ is a "restrictions of a DTM function" $d_{P_X, m_0}$, if there exists a grid $\mathcal{G} = \{x_i\} \subset \mathbb{R}^d$, with $f_X(\sigma_i) = d_{P_X, m_0}(x_i)$.

From Theorem 5.2,

$$\|\mathcal{C}_X - \mathcal{C}_{X'}\|_1 \leq C_K \|f_X - f_{X'}\|_\infty.$$

Now we further bound $\|f_X - f_{X'}\|_\infty$. Note that

$$\|f_X - f_{X'}\|_\infty = \max_{\sigma \in K} |f_X(\sigma) - f_{X'}(\sigma)|$$

$$= \max_{x \in \mathcal{G}} \left| d_{P_X, m_0}(x) - d_{P_{X'}, m_0}(x) \right|$$

$$\leq \left\| d_{P_X, m_0} - d_{P_{X'}, m_0} \right\|_\infty.$$

And from Chazal et al. (2011)[Theorem 3.5],

$$\left\| d_{P_X, m_0} - d_{P_{X'}, m_0} \right\|_\infty \leq \frac{1}{\sqrt{m_0}} W_2(P_X, P_{X'}).$$

Hence putting these things together gives

$$\|\mathcal{C}_X - \mathcal{C}_{X'}\|_1 \leq \frac{C_K}{\sqrt{m_0}} W_2(P_X, P_{X'}).$$

□

*Proof of Theorem 5.4.* By Theorem 5.2, there exists $C_K > 0$ depending only on $K$ such that

$$\|\mathcal{C}_X - \mathcal{C}_{\tilde{X}}\|_1 \leq C_K \| f_X - f_{\tilde{X}} \|_\infty = C_K \max_{\sigma \in K} |\xi(\sigma)|. \tag{11}$$

Since $|\xi(\sigma)| \leq \epsilon/2$ almost surely, then $\|f_X - f_{\tilde{X}}\|_\infty \leq \epsilon/2$ and equation 11 gives the almost-sure bound; the expectation bound follows immediately.

For concentration, note that $\xi(\sigma)$ is a mean-zero random variable with bounded support on $[-\epsilon/2, \epsilon/2]$, it is therefore $\epsilon$-subgaussian. Furthermore, note that we have

$$\mathbb{P}(\|\mathcal{C}_X - \mathcal{C}_{\tilde{X}}\|_1 > \varepsilon) \leq \mathbb{P}\left(C_K \max_{\sigma \in K} |\xi(\sigma)| > \varepsilon\right),$$

by equation 11, and that for any $t > 0$,

$$\mathbb{P}\left(\max_{\sigma \in K} |\xi(\sigma)| > t\right) \leq \sum_{\sigma \in K} \mathbb{P}(|\xi(\sigma)| > t) \leq 2N \exp\left(-\frac{t^2}{2\epsilon^2}\right),$$

by the union bound and the subgaussian tail. Setting $t = \varepsilon/C_K$ yields the stated high-probability inequality.

□

*Proof of Corollary 5.5.* By Proposition 5.1,

$$\|\mathcal{O}_\theta(\tilde{X}) - \mathcal{O}_\theta(X)\|_1 \leq \frac{2L}{\Delta t} \|\mathcal{C}_{\tilde{X}} - \mathcal{C}_X\|_1.$$

Combining this transfer inequality with Theorem 5.4 yields the desired bounds.

□

### F.4 Proofs of Appendix E

*Proof of Theorem E.1.* The proof of Theorem E.1 is in a similar manner from the proof of Wasserstein Stability Theorem of Cohen-Steiner et al. (2010). From equation 6, it is sufficient to show that there exists $C'_K$ depending only on $K$ such that

$$\sum_{k=0}^{\infty} W_1(\mathcal{D}_k(X), \mathcal{D}_k(X')) \leq C'_K d_I(f_X, f_{X'}).$$

Fix $k \geq 0$, and let $\epsilon_k := W_{\infty}(\mathcal{D}_k(X), \mathcal{D}_k(X'))$ be the bottleneck distance between two diagrams $\mathcal{D}_k(X)$ and $\mathcal{D}_k(X')$. Let $\gamma_k : \mathcal{D}_k(X) \to \mathcal{D}_k(X')$ be the bijection that realizes the bottleneck distance, i.e., for any $p \in \mathcal{D}_k(X)$,

$$\|p - \gamma_k(p)\|_{\infty} \leq \epsilon_k.$$

Then 1-Wasserstein distance $W_1(\mathcal{D}_k(X), \mathcal{D}_k(X'))$ satisfies

$$\begin{aligned}
W_1(\mathcal{D}_k(X), \mathcal{D}_k(X')) &= \inf_{\gamma} \sum_{x \in \mathcal{D}_k(X)} \|x - \gamma(x)\|_{\infty} \\
&\leq \sum_{x \in \mathcal{D}_k(X)} \|x - \gamma_k(x)\|_{\infty} \\
&\leq \epsilon_k |\mathcal{D}_k(X)|.
\end{aligned}$$

And hence if we let $\epsilon := \sup_{k \geq 0} \{\epsilon_k\}$, then summing over $k \geq 0$ gives

$$\begin{aligned}
\sum_{k=0}^{\infty} W_1(\mathcal{D}_k(X), \mathcal{D}_k(X')) &\leq \sum_{k=0}^{\infty} \epsilon_k |\mathcal{D}_k(X)| \\
&\leq \epsilon \sum_{k=0}^{\infty} |\mathcal{D}_k(X)|.
\end{aligned}$$

Now $\sum_{k=0}^{\infty} |\mathcal{D}_k(X)|$ is the number of points in persistence diagrams of all homological dimensions on $K$. This can be bounded by some constant $C'_K$ that depends only on $K$: one rough bound can be as $|\{\sigma : \sigma \in K\}|^2$, since each point in persistence diagrams has a unique pair $(\sigma_b, \sigma_d)$ of a birth simplex $\sigma_b$ and a death simplex $\sigma_d$. And therefore,

$$\sum_{k=0}^{\infty} W_1(\mathcal{D}_k(X), \mathcal{D}_k(X')) \leq C'_K \epsilon.$$

Now from $f_X$ and $f_{X'}$ being on a finite simplicial complex $K$, they are $q$-tame (see Section 3.8 from Chazal et al., 2016a). So from the bottleneck stability theorem (see e.g., Section 5.1 and Theorem 5.23 from Chazal et al., 2016a), for all $k \geq 0$,

$$\epsilon_k \leq d_I(f_X, f_{X'}).$$

And hence,

$$\sum_{k=0}^{\infty} W_1(\mathcal{D}_k(X), \mathcal{D}_k(X')) \leq C'_K d_I(f_X, f_{X'}).$$

$\square$

*Proof of Corollary E.2.* From Theorem E.1,

$$\|\mathcal{C}_X - \mathcal{C}_{X'}\|_1 \leq C_K d_I(f_X, f_{X'}).$$

Then from Lemma 4.3 of Chazal et al. (2014a),

$$d_I(f_X, f_{X'}) \leq d_{GH}(X, X').$$

Hence putting these things together gives

$$\|\mathcal{C}_X - \mathcal{C}_{X'}\|_1 \leq C_K d_{GH}(X, X').$$

$\square$

