# OpenReview forum: "ECLayr: A Fast and Robust Topological Layer via Euler Characteristic Curves"
_TMLR — Decision pending for TMLR_

### Review · Reviewer_d5q8 · 2026-03-22

**Summary Of Contributions:**

This paper introduces ECLayr, a topological layer for deep learning built on Euler Characteristic Curves (ECCs) rather than persistent homology (PH). The paper makes six claimed contributions: (1) demonstrating that ECC computation is substantially faster than PH; (2) supporting generic differentiable filtrations across diverse data modalities (images, point clouds, voxels); (3) proposing a novel backpropagation scheme via distributional derivatives that avoids the vanishing gradient problem inherent in sigmoid-based approximations (as used in DECT); (4) providing stability analysis with deterministic, expected, and high-probability bounds under noisy perturbations; (5) demonstrating integration into topological autoencoders; and (6) conducting classification experiments on MNIST, ORBIT5K, and MedMNIST3D datasets showing competitive performance with significant speedups.

**Audience:**

Yes

**Audience Explanation:**

I think the computational bottleneck of persistent homology is a widely recognized barrier to adoption, and many people in the area of topological deep learning will be interested in. A principled, efficient alternative that can be plugged into standard architectures addresses a genuine need. The paper would be of interest to researchers working on TDA integration into deep learning, geometric deep learning practitioners, and those working with high-dimensional data (e.g., 3D medical imaging) where PH is infeasible.

**Broader Impact Concerns:**

The paper is primarily a methodological contribution in topological deep learning and does not raise significant ethical concerns.

**Claims And Evidence:**

Yes

**Claims Explanation:**

The theoretical claims are well-supported: the runtime analysis, gradient non-vanishing guarantees, and stability bounds are rigorously proven with detailed proofs in the appendix. The computational efficiency claims are convincingly demonstrated across multiple dimensions.
However, several empirical claims are less convincingly supported. The claim that ECLayr "achieves performance comparable to state-of-the-art PH-based layers" is somewhat overstated: it outperforms PH-based layers in this specific experimental setup (small training sets, simple architectures), but the paper acknowledges this may be because ECC's simplicity avoids overfitting in low-data regimes, which limits the generalizability of this conclusion. The stability experiments are purely theoretical; no empirical validation of the stability bounds (e.g., measuring actual output deviation under controlled noise levels and comparing to the theoretical predictions) is provided.

**Requested Changes:**

1. Please strengthen the DECT comparison. The authors could either implement the actual DECT method (not just the DECC proxy) on at least one dataset where DECT is applicable (e.g., graph/mesh classification), or provide a thorough discussion of why DECC is a faithful proxy and what limitations this comparison has.
2. Please report the standard metrics such as trustworthiness, continuity, kNN accuracy in the latent space, or topological accuracy (e.g., comparing persistence diagrams of input and latent representations).
3. Please clarify the scope of applicability more precisely. The paper oscillates between claiming broad applicability and acknowledging limitations to "simple topological structures." Provide concrete guidance: what topological complexity (e.g., in terms of Betti numbers or persistence diagram cardinality) marks the boundary where ECLayr becomes inadequate?
4. Also, are there other ways to vectorize ECCs (e.g., basis function decompositions, Fourier coefficients) that might capture more information while maintaining efficiency?
5. Could benefit from more realistic corruption scenarios.

---

> ### Author Response · Authors · 2026-04-30
>
> We thank the reviewer for their valuable comments and suggestions. Below, we address each of the main points raised in the review.
>
> - **RC1:** Thank you for the valuable comment. As the reviewer suggested, we have updated the manuscript to include a discussion in the experimental setup of Section 6.3 explaining why DECC serves as a faithful proxy.
>
> - **RC2:** We thank the reviewer for this constructive suggestion. Quantitative metrics were not initially included because the topological autoencoder experiment was intended primarily as a motivating example of how ECC can be used to constrain the shape of the latent space, rather than to claim superiority over alternative approaches. Nevertheless, we agree that reporting such metrics would provide complementary evidence to support the visual results, and we have revised the manuscript accordingly to include Table 4 in Appendix B.2, which provides quantitative results across multiple evaluation metrics.
>
> - **RC3:** We appreciate this helpful suggestion and welcome the opportunity to clarify the scope of applicability more precisely. First, we respectfully note that our claim of "broad applicability" concerns *data modalities*, in the sense that our framework can accommodate various data types through appropriate choices of filtration (Introduction and Discussion). We also acknowledge its inherent limitations with respect to *expressiveness*, which arise from the tradeoff between ECC- and PH-based approaches (Section 2.2). Accordingly, we do not claim that ECLayr generally outperforms PH-based methods. Rather, we emphasize that ECLayr and PH-based approaches offer distinct advantages, and we aim to characterize the settings in which ECLayr is preferable. Thus, we have revised Section 2.2 in the updated manuscript to present a more clear guidance of such settings.
>
> - **RC4:** This is a great point. We believe alternative vectorization methods, such as the basis function decomposition suggested by the reviewer, may be also be feasible. However, we chose uniform grid sampling because it is the most direct and computationally simple way to obtain a fixed-dimensional representation of the ECC. It retains direct interpretability and integrates naturally with our proposed backpropagation scheme. In particular, the sampled-ECC representation preserves the step-function structure of the ECC, yields a transparent O(N+v) implementation, and avoids introducing an additional approximation layer whose effect would need to be analyzed separately. Nevertheless, exploring alternative ECC embeddings would be an interesting direction for future work.
>
> - **RC5:** We thank the reviewer for this suggestion. We agree that the corruption scenario in Section 6.3 is simplified and may not fully reflect the variety of real-world perturbations encountered in practice. As stated in the manuscript, our intention in this experiment was to conduct a controlled and systematic stress test rather than to exhaustively model real-world corruptions. Nevertheless, we expect the qualitative robustness conclusions to extend beyond this particular setup, since the stability analysis in Section 5 provides guarantees at the level of perturbations to the filtration and is therefore not tied to a specific corruption mechanism. Extending the experiments to more realistic  and application-specific corruption settings would be a meaningful direction for future work.
>
> We again thank the reviewer for their time and input, and hope that our response has sufficiently addressed your questions. We would be pleased to provide further clarification if any concerns remain.

---

> ### Comment · Reviewer_d5q8 · 2026-05-16
>
> I thank the authors for their detailed response and for the revisions made to the manuscript. Below I share my reactions to each point.
>
> RC1. I appreciate the added discussion in Section 6.3 explaining why DECC serves as a faithful proxy for DECT. This is a reasonable resolution given the scope of the paper, though I would encourage the authors to ensure the discussion explicitly notes any limitations of the proxy (e.g., settings where DECC and DECT might diverge in behavior) so readers can interpret the comparison appropriately.
>
> RC2. I appreciate the addition of Table 4 in Appendix B.2 reporting quantitative metrics for the topological autoencoder experiment. This addresses my concern, assuming the table includes standard latent-space evaluation metrics such as trustworthiness, continuity, or kNN accuracy, and/or a topological measure (e.g., persistence diagram comparison between input and latent representations).
>
> RC3. I appreciate the clarification that "broad applicability" refers to data modalities rather than topological expressiveness, and the revision of Section 2.2. However, my original concern remains partially open: I had asked for concrete guidance on what topological complexity marks the boundary where ECLayr becomes inadequate (e.g., in terms of Betti number ranges, persistence diagram cardinality, or empirically observable signals). The reframing helps clarify intent, but practitioners deciding whether to use ECLayr would still benefit from a more quantitative criterion, even if approximate or empirical. If a precise boundary is difficult to establish, a brief qualitative rule of thumb supported by an illustrative example would already be valuable.
>
> RC4. I accept the authors' justification for choosing uniform grid sampling: the design choice is well-motivated by simplicity, interpretability, and compatibility with the proposed backpropagation scheme. Treating alternative vectorizations as future work is reasonable.
>
> RC5. I understand the rationale for the controlled stress-test design and agree that the theoretical guarantees in Section 5 are filtration-level and thus mechanism-agnostic. That said, my original review also noted that no empirical validation of the stability bounds was provided (e.g., comparing measured output deviation under controlled noise to theoretical predictions). Even a small-scale empirical check confirming that the bounds are non-vacuous and predictive in practice would meaningfully strengthen the stability claims. I would encourage the authors to consider adding such a sanity check, even in the appendix.
>
> Overall, the revisions improve the paper, and most of my concerns are now adequately addressed. The remaining items I would most like to see strengthened are the concrete scope guidance (RC3) and a light empirical validation of the stability bounds (RC5).

---

### Review · Reviewer_EqBJ · 2026-03-24

**Summary Of Contributions:**

The paper proposes ECLayr, a differentiable topological layer based on the Euler Characteristic Curve (ECC) as a computationally cheaper alternative to persistent-homology-based topological layers. The main idea is to replace PH summaries with ECC summaries, derive a practical differentiable approximation for backpropagation, and demonstrate the resulting layer on classification and topological autoencoder tasks.

Strengths
1. The computational motivation is clear: ECC is significantly cheaper than PH and therefore potentially more usable in practice.
2. The paper is generally well written.

Weaknesses
1. The theoretical contribution is more modest than the paper sometimes suggests. The counting-ECC construction is natural, and the derivative construction feels more like a practical differentiability patch than a major conceptual advance.
2. The theory mostly addresses perturbation stability of the ECC of a single object, whereas the more relevant classification question is whether topologically similar but distinct objects are mapped to similar useful representations.
3. The empirical results show utility, but do not convincingly demonstrate how a differentiable ECC layer is needed and whether the success is based on the correct topology capturing of the ECC layer.
4. The comparison with DECT is somewhat limited. The strongest distinction is in the gradient computation and stability guarantee. But an ablation on the gradient method is missing and the stability results are mostly bookkeeping with coarse L-infty type results for two filtrations on the same space.

**Additional Comments:**

N/A

**Audience:**

Yes

**Audience Explanation:**

Yes, there is some interest in incorporating topology into deep learning.

**Claims And Evidence:**

No

**Claims Explanation:**

The empirical results show that ECLayr can improve performance, but they do not clearly establish why. In particular, the paper does not show that the ECC branch is actually capturing topology-relevant class structure, rather than merely contributing a useful auxiliary global descriptor.

**Requested Changes:**

Strengthen the empirical link between ECC and tasks. The paper should provide evidence that the ECC branch improves performance in a topology-relevant way and that a learnable layer is necessary rather than relying solely on data augmentation. For example, how does the ECC look in the case of different objects in the same class? How does the ECC look for the noise-contaminated case?

---

> ### Author Response · Authors · 2026-04-30
>
> We thank the reviewer for the constructive feedback and address the major concerns below.
>
> - **W1 \& W4:** We acknowledge that the primary contribution of our work is best understood as a principled and practically useful layer construction, rather than as a conceptual advancement. Nevertheless, we believe it still may constitute a meaningful contribution, as it offers a practical alternative to address an important problem in TDA: computational bottleneck. Our proposed model notably improves runtime over previous approaches by directly targeting the gradient of the indicator function, rather than relying on a smooth approximation of the indicator function, while remaining fairly general to be applicable across a range of data modalities. Moreover, we provide theoretical guarantees for the output of our model, establishing several stability results that include deterministic, expected, and high-probability bounds. The deterministic stability guarantees further extend to the case of two filtrations defined on different simplicial complexes (Appendix E), and to empirical distributions of the input (Corollary 5.3). We also present a theoretical gradient comparison analysis against DECT, evaluating the approximated gradient's consistency to the true underlying gradient (previously Appendix E, moved to Section 4 in updated manuscript). We hope this clarification helps convey our intended contribution more accurately, and we would be grateful for the reviewer’s consideration of the practical and theoretical value of the proposed approach.
>
> - **W2, W3 \& Requested Changes**: Thank you for this helpful suggestion. We agree that demonstrating how the ECCs captured by ECLayr connect to downstream tasks is important, and we believe that clarifying this connection would strengthen the manuscript. To this end, we have added several analyses to Appendix B.6 for different noise levels of the MNIST data, including (i) visualizations of the average ECC for each class, (ii) heatmaps of the pairwise distance matrices among class-wise average ECCs, and (iii) a table summarizing the top-3 closest classes for each class.
>
> We hope that our responses sufficiently resolve your concerns, and again express our gratitude for the reviewer's time and input. If any questions remain, we would be pleased to provide further clarification.

---

### Review · Reviewer_Tqpa · 2026-04-16

**Summary Of Contributions:**

**Summary**
This paper suggests a new topological layer, ECLayr, for learnable networks. It is based on the Euler characteristic curve (ECCs), a coarser topological feature than persistent homology. Conversely, ECCs are much quicker to compute. To back-propagate through the layer, the paper proposes a new approximation inspired by distributional derivatives, which achieves better approximation than prior work using sigmoids. The authors analyze the run time complexity and stability of ECLayr in detail theoretically and provide an empirical comparison to other topological learning approaches.


**Strengths**

- S1: The method seems novel and the presentation is for the most part easy to follow.
- S2: The speed improvement over DECC seems drastic.
- S3: The performance improvements are modest, but paired with the speed-up, useful.
- S4: The extensive theory sections provide a decent understanding of properties of the ECLayr  method.
- S5: The theory seems correct up to some questions, see below.

**Weaknesses**

*Major*
- Randomness in Prop 3.1. The randomness in Prop 3.1 is not clear at all. At first glance everything seems deterministic.

- Independence in Prop 3.1.: Why should the independence assumption in (iii) hold? Especially, indicators of simplices that are contained in each other, are likely not independent.

- Desiderata in Sec 4: I found it hard to appreciate Sec 4 without Appendix E. For instance, from section 4 it does not become clear what the "true derivatives" should even be. Moreover, it would be good to clarify in Prop 4.1. that $\lambda \to \infty$ causes the last statement. Without Appendix E, it is also not clear why one would even what to send $\lambda \to \infty$. Similarly, just from Sec 4.2. it does not become clear why one does not similarly need to send $\beta \to \infty$. Much of this gets clarified in Appendix E, but I would strongly suggest to refactor to convey the necessary intuition already in the main paper.

- Independence assumption in Thm 5.4: It seems that the proof of Thm 5.4 (first $\leq$ in Eq (11)) relies on $f_\tilde{X}$ being a filtration function itself, which requires monotonicity. But how can the $\xi(\sigma)$'s be independent if they are required not to destroy $f_X$'s monotonicity?

- Contextualization of Sec 5. It would be very helpful if the theoretical results of Sec 5 would be contextualized more. What do we gain from the established bounds? Are the assumptions plausible? Can we find clear evidence of these theorems in the experiments or, if not, could one devise experiments to showcase the bounds empirically?

- Performance gain vs run time: While there is a nice performance boost on MNIST and ORBIT5K essentially without increased runtime (ECLayr(i) setting), the improvements on MedMNIST3D are much more modest and require a more substantial increase in runtime.


*Minor*
- Notation $\mathcal{E}(t_i)$ is slightly confusing as $\mathcal{E}: \\{1, \dots, v\\} \to \mathbb{R}$.
- Clarify what is mean by "parameter" in Theorm 5.4.
- Why did the authors not compare their method against the method of Carlsson and Gabrielsson 2020?
- For context, it would be nice to reproduce the result of the original TopoAE in Fig 4.
- For a fair comparison, Figure 5 and Table 2 should perhaps include CNN + DECC(i), i.e., DECC without backprop. If I understand correctly, DECC is only slower than ECLayr in the backward pass.
- Table 3 shows many more decimal places than the value of the standard deviation. I would recommend to only show one more than the value of the standard deviation (e.g. 2 decimal places if the std is $\geq 0.1$).
- While the authors give a qualitative example of what PH but not ECC can distinguish, it would be interesting to find a real-world setting where PH outperforms ECCs so that users get a feeling for the level of complexity that requires which topological descriptor.


**Questions**
- Q1: The main innovation of the paper hinges on the use of a fixed grid. Backpropagation through PH typically directly uses the critical filtration values (+ something Persistent Images to achieve fixed dimensional output) and thus sidesteps this issue. Why is such a solution not possible for ECCs?

- Q2: I am quite surprised that the runtime of DECC is so much worse than that of ECLayr, especially when embedded into a ResNet architecture. If I understand correctly, the main reason for better runtime is that ECLayr avoids $v$ evaluations of $\exp$. But this should be dwarfed by the cost of the forward pass of the remaining ResNet. Similarly, I am surprised that methods based on full PH actually are comparable in run time to DECC (Table 1).

- Q3: How is robustness at the end of Sec 6.3 defined? I do not see an obvious trend in Fig 5.

- Q4: How were the values of $\lambda$ in DECC determined? And why do these values provide a good trade-off between the Eq (8) and (9)?

- Q5: In the proof of Prop 5.1: Why can we expand $\|\mathcal{C}_X(tseq) - ...\|_1$ as claimed? I would have expected only a $\leq$ since $| \sum a_i ...|\leq \sum |a_i|...$. Similarly, for $\| \mathcal{C}_X - ...\|_1$. Moreover, I do not get the expansion of the latter in the first place.

**Typos**
"minimized" --> "maximized " in Proof of Prop E.1.

**Audience:**

Yes

**Audience Explanation:**

Yes. Topological methods are still gaining popularity and especially the speed-up offered by ECLayr seems useful.

**Claims And Evidence:**

Yes

**Claims Explanation:**

Mostly yes. Empirical results are rather slim and I have some questions regarding the theory, see above.

**Requested Changes:**

See weaknesses above. In particular,
- Clarify questions on the assumptions of the statements and on the proofs.
- Contextualize the results of Sec 4 and 5 better.
- Insert direct cross links from a theorem to its proof or at least reproduce the theorem before its proof in the appendix.

---

> ### Author Response · Authors · 2026-04-30
>
> Thank you for taking your time and effort to review our work. Below, we address the main points raised in the review.
>
> - **Prop 3.1:** We appreciate the valuable comments and have revised the manuscript to clarify the source of randomness in Prop 3.1. We also agree that the independence assumption in Prop. 3.1 (iii) is generally unnatural for monotone filtrations. We had initially overlooked this point and are grateful to the reviewer for bringing it to our attention. Since our main runtime message is already captured by the deterministic and expected bound in (i) and (ii), we removed the concentration statement to avoid imposing an unrealistic probabilistic model.
>
> - **Sec 4 \& Sec 5:** We thank the reviewer for this helpful observation. We agree that clarifying the propositions in Sec. 4 and integrating the material from Appendix E would enhance the presentation of that section, and we have made the necessary revisions in the updated manuscript. Furthermore, to better contextualize Sec. 5, we have added explanations clarifying the interpretation and implications of each stability bound.
>
> - **Thm 5.4:** We agree that the monotonicity requirement on $f_{\tilde X}$ restricts the independence of $\xi(\sigma)$s, and we sincerely thank the reviewer for raising this point. We have now updated Thm 5.4 to consider bounded perturbations that do not effect the monotonicity of $f_{\tilde X}$.
>
> - **Performance vs. Runtime:** Thank you for raising this point. We would like to respectfully note that the increased runtime of ResNet+ECLayr relative to ResNet on MedMNIST3D is partly attributable to several external factors.
>     1. Limited computational resources: As in the original implementations of prior PH-based topological layers, ECLayr also runs on CPU because the GUDHI package used to compute filtration values relies on CPU-based execution. Thus, ResNet+ECLayr involves both GPU-based neural network computation and CPU-based topological layer computation, whereas the baseline ResNet can be run entirely on the GPU. Our GPU server, however, is equipped with a relatively low-performance CPU that is primarily intended for loading data rather than intensive computation. Such limitation in CPU performance becomes more pronounced as the data dimensionality increases, leading to a larger runtime gap between ResNet+ECLayr and ResNet on MedMNIST3D. We believe that the runtime gap would likely have been much smaller with a more capable CPU.
>     2. In our current implementation, the CPU and GPU computations for ResNet+ECLayr are not executed in parallel via multiprocessing. Synchronizing these operations to run simultaneously could partially reduce runtime of ResNet+ECLayr.
>
>   The runtime gap observed on MedMNIST3D is partly driven by the external factors discussed above, rather than by our proposed method itself, and could be narrowed by addressing them.
>
> - **Q1:** As the reviewer has correctly pointed out, backpropagation through PH directly uses critical filtration values without necessitating a fixed grid. However, when backpropagating through a *functional Hilbert space mapping* of PH, such as persistence landscapes, a fixed grid is required to represent the functional summary and backpropagate the upstream gradient through the grid points (e.g., Theorem 3.1. in [1]). Similarly, since ECC is a functional summary defined as an alternating sum of indicator functions, the use of a fixed grid is unavoidable, and our approach provides an efficient method for backpropagation in this setting.

---

> ### Author Response · Authors · 2026-04-30
> **Cont'd**
>
> - **Q2 \& omission of CNN+DECC(i):** This is a great point. The primary reason for the significant runtime difference is that ECLayr improves computation over DECC in *both* the forward and backward passes (Section 4.1). DECC's sigmoid approximation requires $v$ evaluations of the sigmoid function in the forward pass and $v$ evaluations of its gradient in the backward pass, whereas ECLayr requires evaluation at only a single grid point in the forward pass and at only two grid points in the backward pass. The difference in the forward pass computation causes CNN+DECC(i) to be much slower than CNN+ECLayr(i), while yielding the same output, as a very large $\lambda$ is used to obtain a tight sigmoid approximation when backpropagation is not required. We therefore did not consider it informative to include CNN+DECC(i) in the comparison. Moreover, when integrated into ResNet, the runtime of DECC was not dwarfed by the remaining ResNet computation since, as previously discussed, the ResNet and ECLayr/DECC computations were not parallelized for synchronized execution. Regarding the runtime comparison between DECC and PH, DECC demonstrates better scalability than PH, with the runtime gap becoming more pronounced as the data dimensionality increases (Table 1). The comparable runtime observed on MNIST is likely due to the relatively modest dimensionality of the dataset, coupled with implementation-level differences. In particular, PH computation is performed entirely within the GUDHI package, which may have a more computationally optimized implementation, whereas ECLayr/DECC uses GUDHI only to compute filtration values. Nevertheless, converting PH into alternative representations suitable for downstream models introduces additional computational overhead, resulting in a more notable runtime gap between DECC-based models and PH-based models (Table 2).
>
> - **Q3:** Here, robustness refers to resistance against noise and corruption. Thus, in Figure 5, we aim to examine how the performance gap between CNN+ECLayr and the baseline CNN evolves as the noise level increases. On MNIST, the performance of CNN+ECLayr degrades more slowly than the baseline CNN up to $10\sim15$% noise level, while on ORBIT5K, it consistently exhibits slower performance degradation across the tested noise levels.
>
> - **Q4:** This is a great question! As the original DECT paper does not provide explicit guidance on how to choose $\lambda$, we adopted one of the values used in their original implementation.
>
> - **Q5:** Thank you for pointing this out! We apologize as the notation in the proof of Prop. 5.1 is somewhat confusing and the proof was written too compactly. In the expansion of $\Vert \mathcal{C}_X(tseq) - \mathcal{C}_X(tseq)\Vert_1$ and $\Vert \mathcal{C}_X - \mathcal{C}_X\Vert_1$, the reviewer is correct in noting that, in general, only the inequality $\leq$ would hold. However, the expansion is possible in the current setting because the intervals on which the indicator functions are defined on are mutually disjoint. We have revised Appendix F.3 to provide additional explanation and thereby make the proof clearer.
> \end{itemize}
>
> - We also addressed many of the other minor comments in the updated manuscript, and we are grateful to the reviewer for the helpful suggestions.
>
> [1] K. Kim et al. Pllay: Efficient
> topological layer based on persistent landscapes. NeurIPS, 2020.
>
> We again thank the reviewer for the constructive feedback. We wish that our response has adequately addressed your questions and concerns, and we would be more than happy to provide further clarification if needed.

---

> > ### Comment · Reviewer_Tqpa · 2026-05-15
> >
> > Dear authors,
> >
> > many thanks for the reply and the revised and much improved document. Many of my issues got resolved. I have a few follow-up questions:
> >
> > **Thm 5.4:** The condition $max( ....) \leq \varepsilon$ should be $min(...) \geq \varepsilon$, right? You mention that for typical cubical complexes this assumption does not hold, making the theorem non-applicable. I think it also does not hold for many typical simplicial complexes, such as Vietoris-Rips or other flag complexes. Please clarify which popular complexes the theorem does / does not apply to and argue for its relevance given the limited applicability.
> >
> > **Q3:** On Orbit5K, I see the difference in robustness. On MNIST, I find it very hard to spot the trend up to 10/15% noise level. If I had to specify any difference, I would even say that CNN+EClayr deteriorates worse than the pure CNN over the full noise range on MNIST.
> >
> > **Q4:** Please give more details on how you chose $\lambda$ exactly. Did you check that lower $\lambda$ values do not trade-off speed and performance more favorably for DECC?
> >
> > **Q5 (Prop 5.1):** Why do we need to assume the existence of a suitable $\Delta t$? Does it not always exist?
> >
> > Moreover, some points from my original review have not yet been addressed, e.g.
> > - **Topo AE** result in Fig 4 still missing, also in new Table 4
> > - **Small performance boost on MedMNIST3D despite large increase in runtime.** Would e.g. training the ResNet 2-3 times longer (matching the wall time of ResNet + ECLayr) close the performance gap to ResNet + ECLayr?

---

### Author Response · Authors · 2026-04-30

We are deeply grateful to the reviewers for their careful evaluation and constructive feedback, and we sincerely apologize for the delay in our response. Based on the reviewers' valuable comments and feedback, we have made the corresponding revisions to our manuscript. **All revisions are marked in red** so that they can be easily located. We also note that all references to our paper (such as Sections, etc.) in the responses correspond to the **newly uploaded revised version**.